# Seasonality in Terminus Ablation Rates for the Glaciers in Greenland (Kalaallit Nunaat)

Aman KC[1], Ellyn M. Enderlin[1], Dominik Fahrner[2], Twila Moon[3], and Dustin Carroll[4,5]

[1]Department of Geosciences, Boise State University
[2]Department of Glaciology and Climate, Geological Survey of Denmark and Greenland
[3]National Snow and Ice Data Center, Cooperative Institute for Research in Environmental Sciences, University of Colorado Boulder
[4]Moss Landing Marine Laboratories, San José State University
[5]Jet Propulsion Laboratory, California Institute of Technology

**Correspondence:** Aman KC (amankc@u.boisestate.edu)

**Abstract.**

Marine-terminating glaciers of Greenland (Kalaallit Nunaat) have undergone accelerated mass loss since the 1990s, with a substantial portion due to the effects of dynamic change. Conventional assessments of dynamic mass loss, however, often ignore the influence of terminus advance or retreat on the timing of mass loss. Here we construct and analyze a decade (2013–2023) of monthly ice flux driven both by temporal variability in ice flow (i.e., discharge) and terminus position change — collectively called terminus ablation — for 49 marine-terminating glaciers in Greenland. We calculate terminus ablation rates using open-source datasets, including terminus position, ice surface elevation, ice thickness, and glacier speed. For the majority of glaciers, we observe coincident seasonal variations in terminus position and discharge. However, seasonal variations in terminus ablation are much larger than those in discharge. For the northwest and central west sectors, where the highest fraction of outlet glaciers is included in our terminus ablation dataset, terminus ablation varies by ∼51 Gt/yr and ∼25 Gt/yr, respectively, over each year. In contrast, the corresponding variation in discharge is only ∼5 Gt/yr. While our terminus ablation time series do not include every outlet glacier, they suggest that terminus position change is the dominant contributor to Greenland glacier dynamic mass loss at seasonal time scales, in contrast with the relatively small influence of terminus change on decadal-scale mass loss. Since seasonality in mass loss can influence the fate of freshwater fluxes, we recommend that studies concerned with the impacts of Greenland mass loss on local-to-global ocean properties should account for seasonal terminus position change.

## 1 Introduction

Marine-terminating glaciers of Greenland (Kalaallit Nunaat) have been losing mass since the 1990s (Bamber et al., 2018b; Khan et al., 2015; Bollen et al., 2022; Noël et al., 2017), with approximately half of the mass loss due to ice-flow acceleration (i.e., dynamic mass loss) that has been largely triggered by terminus retreat (King et al., 2020; Fahrner et al., 2021). Observations of large-scale seasonal (Moon and Joughin, 2008; Black and Joughin, 2022) and interannual (Carr et al., 2013; Howat et al., 2010; Catania et al., 2018) terminus retreat suggests that the timing and location of freshwater flux entering the

surrounding ocean basins from solid ice discharge, such as iceberg calving, may differ from flux gate-based estimates. Even though terminus retreat has been nearly ubiquitous over the last ∼3 decades (Murray et al., 2015; King et al., 2020; Wood et al., 2021), most estimates of dynamic mass loss only consider changes in speed and thickness at an inland flux gate (Van Den Broeke et al., 2016; Mankoff et al., 2021) and do not account for terminus advance or retreat. Decadal estimates of terminus ablation (i.e., mass loss due to changes in ice flux and terminus position change) indicate that the failure to account for terminus change in dynamic mass loss estimates has resulted in a ∼10% underestimation of freshwater fluxes from Greenland in recent decades (Kochtitzky et al., 2023; Greene et al., 2024). In addition, temporally-detailed but spatially-limited terminus ablation time series suggest that terminus ablation can vary tremendously over seasonal-to-interannual timescales (Porter et al., 2018; Kochtitzky and Copland, 2022), indicating that the precise timing of freshwater fluxes associated with dynamic mass loss may differ from flux-gate-based estimates.

Freshwater fluxes from the Greenland Ice Sheet can have significant impacts on local nutrient availability and marine productivity (Laidre et al., 2008; Hopwood et al., 2018, 2020; Oliver et al., 2023), regional ocean properties (Böning et al., 2016; Cape et al., 2018), and potentially global-ocean circulation (Lenaerts et al., 2015). Local changes in fjord hydrography, circulation, nutrient concentrations, and light availability due to the addition of freshwater can impact the transport of nutrients, life cycles of higher-tropic marine organisms such as fish (Duplisea et al., 2021), timing of phytoplankton blooms (Manizza et al., 2023), hatching success of ichthyoplankton (Bouchard et al., 2020), coastal flooding (Nicholls, 2004), population dynamics of seals, narwhals, and polar bears (Womble et al., 2021; Laidre et al., 2022), and Indigenous people's traditional practices (Hamilton et al., 2000). Recent Greenland mass loss has substantially contributed to upper-ocean cooling and freshening in the North Atlantic Ocean (Robson et al., 2016; Dukhovskoy et al., 2019) and alterations in density gradients through freshwater redistribution, which in turn influences the Arctic Ocean Oscillation (Bamber et al., 2018a; Proshutinsky et al., 2015; Bamber et al., 2012). Because freshwater flux pathways vary seasonally (Luo et al., 2016), precisely when ice mass is transferred to the oceans can influence the spatial distribution of the aforementioned impacts.

Here, we address the need for temporally-resolved, spatially-comprehensive terminus ablation in Greenland by creating and analyzing monthly terminus ablation time series for 49 marine-terminating outlet glaciers distributed around the ice sheet. Terminus ablation time series are calculated using a variety of open-source datasets, which allow us to explore intra-annual and regional patterns in terminus ablation relative to conventional flux-gate-based discharge estimates over a decade (2013–2023).

## 2  Data and Methods

We calculated terminus ablation rates for 49 glaciers spanning the Greenland Ice Sheet (Fig. 1), with the majority of systems located in the central west and northwest sectors. Glaciers were grouped into regions following Mouginot and Rignot (2019) to examine the data for potential regional similarities, including 22 glaciers in the northwest (NW), 12 in the central west (CW), 3 in the southwest (SW), 7 in the southeast (SE), 4 in the central east (CE), and 1 in the northeast (NE). For glaciers with identical names, we distinguished them based on latitude. For example, there were four glaciers named Sermeq Kujalleq, which were labeled N, N1, N2, and N3, with numbering increasing from south to north. The regional distribution of selected glaciers

reflects both the actual concentration of marine-terminating glaciers around the ice sheet and data availability. The primary data limitation was the availability of observation-constrained bed elevations. We included glaciers with bed elevations from BedMachine 5 Greenland (Morlighem et al., 2022) that were either measured directly (using airborne radar sounders or ice-penetrating radar) or derived from mass conservation, with typical uncertainties of <150 m. We did not include glaciers above $\sim$80°N because they have sparse terminus position records due to limited satellite/aerial data availability (Goliber et al., 2022), preventing the construction of the sub-annual terminus position time series that were needed to estimate seasonal terminus ablation rates.

The datasets and methodologies used to construct the terminus ablation time series are illustrated in Fig. 2 and described in detail below. Nearly the same glaciers were analyzed as is Fahrner et al. (2025) and a similar processing pipeline was applied but with several differences in the creation of terminus mass change time series. Briefly, terminus area change was calculated using the method similar to the "curvilinear box method" (Lea et al., 2014) applied to the TermPicks and CALFIN (Calving Front Machine) terminus time series for each glacier and additional manual delineations from 2020–2023. Following a similar approach as Fahrner et al. (2025), terminus delineations were filtered then terminus area time series were converted to mass change estimates using BedMachine, ArcticDEM, and AERODEM elevations. Finally, to reduce potential errors caused by the inclusion of multiple terminus datasets, we combined month-averaged terminus mass change and discharge time series to yield monthly time series of terminus ablation rate.

## 2.1 Terminus change

Glacier terminus positions were obtained from two primary sources: TermPicks (Goliber et al., 2022) and CALFIN (Cheng et al., 2021). TermPicks is a comprehensive compilation of manually-delineated terminus traces gathered from a variety of studies. The dataset integrates terminus traces from a diverse range of image sources, including 11 satellite-based multispectral image datasets, 7 synthetic aperture radar (SAR) datasets, and aerial photographs from 5 different missions. The CALFIN dataset was generated with an automated method that uses neural networks to extract calving fronts from Landsat images. While some terminus traces date back to 1916, the majority of the data occur after 1999, when an increase in satellite coverage enabled the construction of more-temporally-dense time series. The temporal coverage and resolution of terminus position data vary substantially across glaciers. For example, Helheim Glacier is one of the most studied sites, with $\sim$2250 terminus traces, while Saqqarliup Sermia on the west coast has only $\sim$400 terminus traces. To extend the temporal coverage of the terminus trace datasets, monthly manually-digitized terminus traces from 2020–2023 were added to the terminus TermPicks time series. A total of 1,812 new terminus traces are publicly available through the Arctic Data Center (KC et al., 2024b). Terminus traces were digitized in true-color Sentinel-2 and Landsat 8/9 images using GEEDiT (Lea, 2018) following the recommended procedures of Goliber et al. (2022). For each month with solar illumination, the cloud-free terminus image with the acquisition date closest to the middle day of each month was used to trace the terminus, with no preference given to the choice of satellite platform.

Terminus mass change was first estimated by computing terminus area using the approach similar to the curvilinear box method (Lea et al., 2014), then converting terminus area to volume. Fig. 3 describes the processing workflow for the terminus

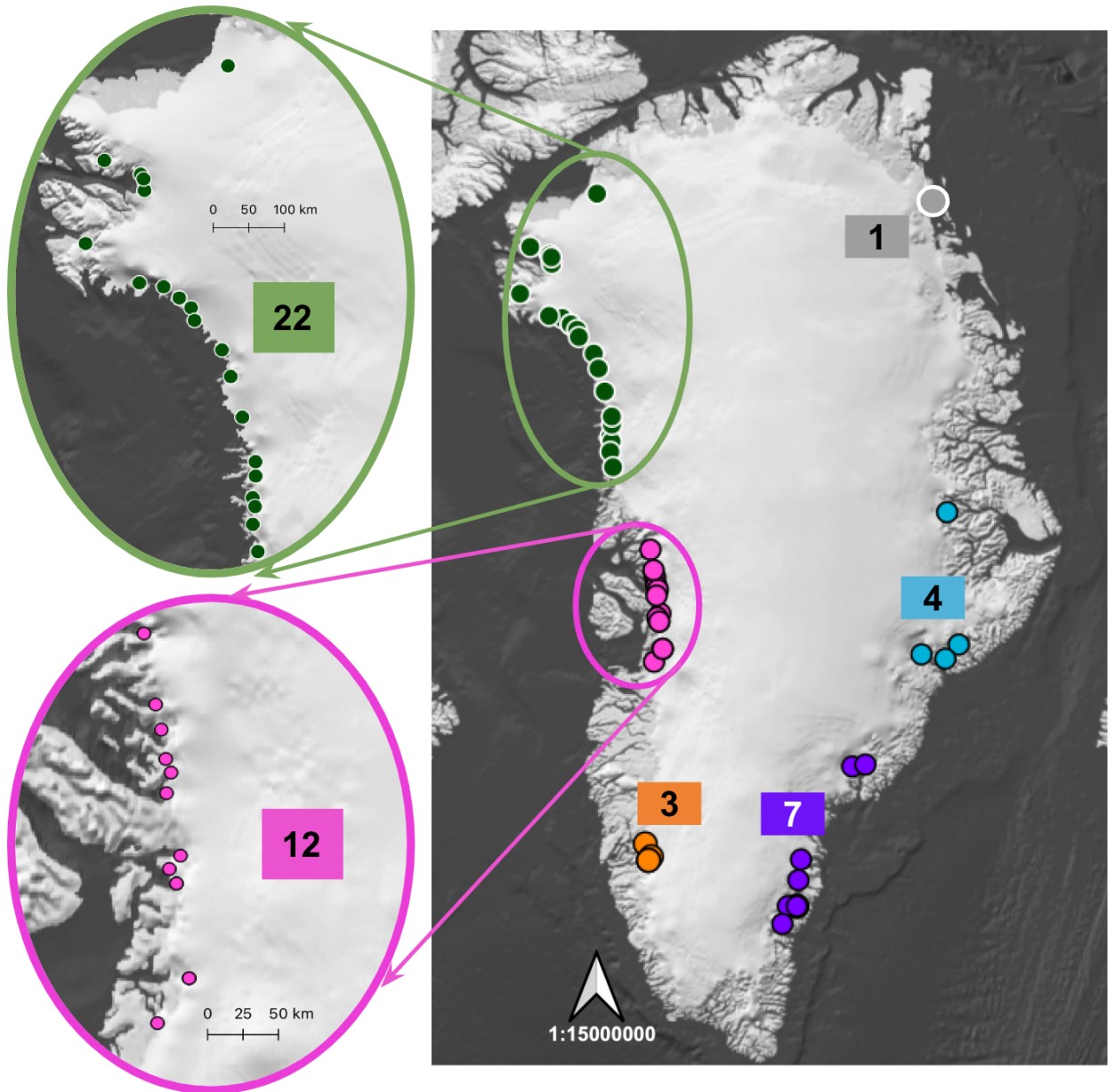

**Figure 1.** Location of the 49 glaciers used in the study. Symbol colors distinguish regions: blue = central east (CE), purple = southeast (SE), orange = southwest (SW), pink = central west (CE), green = northwest (NW), and gray = northeast (NE). The numbers in the colored boxes represent the number of glaciers in the region. Base map is from Natural Earth produced with QGreenland (Fisher et al., 2023; Moon et al., 2023)

.

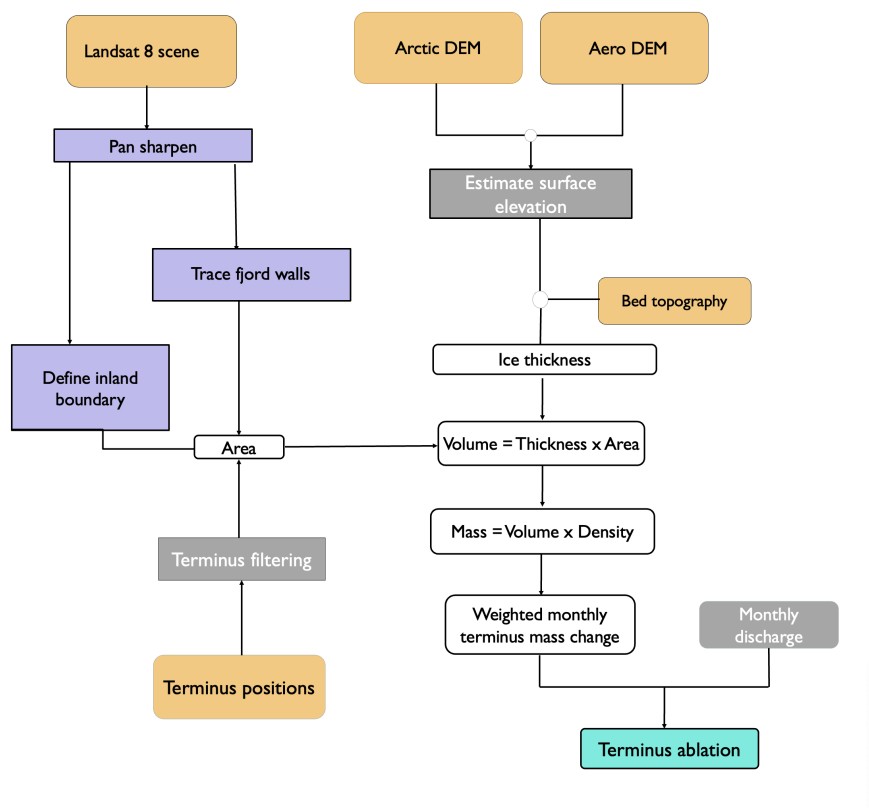

**Figure 2.** Flowchart describing the methodology used to calculate terminus ablation. Rounded rectangles show inputs, intermediate outputs, and final output. Angular gray rectangles show the mathematical processes and purple rectangles show the processes applied to satellite images. Orange rectangles show data from external sources.

data, including standardizing and filtering terminus datasets to minimize effects from the inconsistencies illustrated in Fig. 4. The curvilinear box method for calculating terminus area change relies on terminus traces that span the glacier width to define the terminus area. For each site, a Landsat 8 panchromatic image was used to trace fjord boundaries on either side of the glacier and define the polygon that encompassed the full range of terminus positions from 2013–2023 (Fig. S1). The TermPicks and CALFIN terminus positions were clipped to the fjord boundaries. Terminus traces that were <90% of the minimum fjord width were removed and terminus traces >90% of the minimum fjord width but that did not intersect fjord boundaries were extrapolated to the nearest fjord boundary point.

The composite TermPicks and CALFIN full-width terminus trace time series were then filtered to eliminate duplicate polygons for each date based on accuracy. TermPicks traces were compiled from multiple studies, resulting in varying accuracy due to differences in (1) the methodology used, such as the box (Bunce et al., 2018; Catania et al., 2018) or full-width (Bevan et al., 2012; Zhang et al., 2021) methods, and (2) the image source, with particularly large differences in spatial resolution and overall quality between satellite images (Bunce et al., 2018) and suborbital photographs (Korsgaard, 2021). A detailed description of

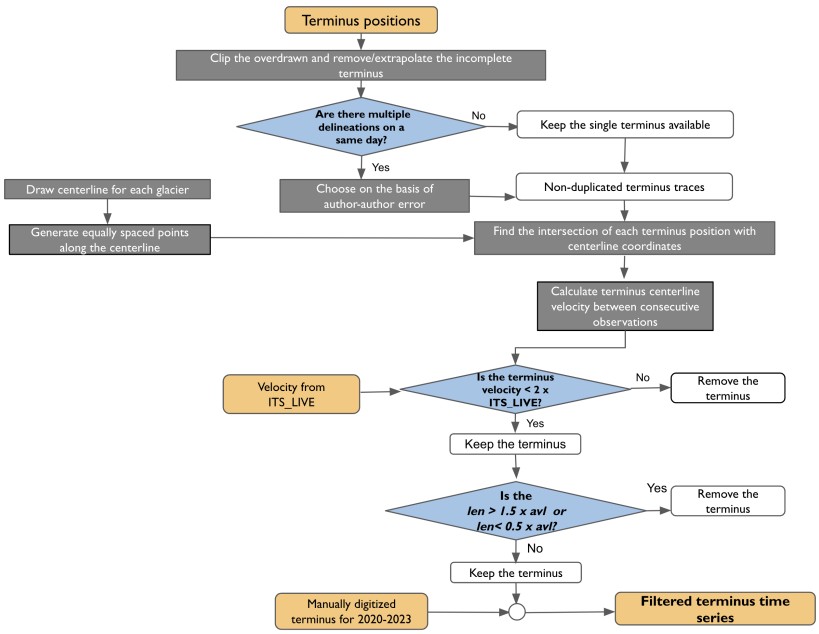

**Figure 3.** Flowchart describing methods for filtering terminus delineations. Orange rounded rectangles show inputs, intermediate outputs, and final output. Gray rectangles show the mathematical processes and blue diamonds show the conditional checks. Here, len = length of individual terminus trace and avl = average length of all the available termini traces.

differences in terminus traces can be found in Goliber et al. (2022) and inconsistencies in the composite terminus datasets are generalized in Fig. 4. Although there can be differences in the manual traces across studies, we consider the manual traces to be more accurate that the automated CALFIN dataset because CALFIN results are prone to error when shadows or thick ice mélange are present near the terminus (Goliber et al., 2022). When multiple traces were available for the same day, the most accurate trace was selected using the "Quality-Flag" attributes from the TermPicks dataset, prioritizing manual traces over the CALFIN data. Although the quality-based filtering removed redundant terminus traces, an additional filtering step was required to eliminate apparent jumps or dips in terminus position that resulted from delineation errors or were not physically possible based on the glacier flow speed (Fig. 4 c,d)(Dryak and Enderlin, 2020; Liu et al., 2021). The rate of centerline terminus position change between consecutive traces was compared with the 2013–2022 time-averaged NASA ITS_LIVE speed (Gardner et al., 2019). If the apparent terminus advance rate was greater than two times the time-averaged speed, the advanced terminus trace was removed. The filtered, full-width terminus traces were then compiled to create a terminus area time series for each glacier (Fig. S1).

Terminus area time series were next converted to terminus volume time series. Surface elevations from ArcticDEM strips (Porter et al., 2022) and the AERODEM (Korsgaard, 2021) and bed elevations from BedMachine 5 (Morlighem et al., 2022) were used to estimate the glacier thickness within each terminus area polygon. Although the ice thickness can vary over a range of timescales, we use only the AERODEM and ArcticDEM to capture long-term changes in ice thickness because they

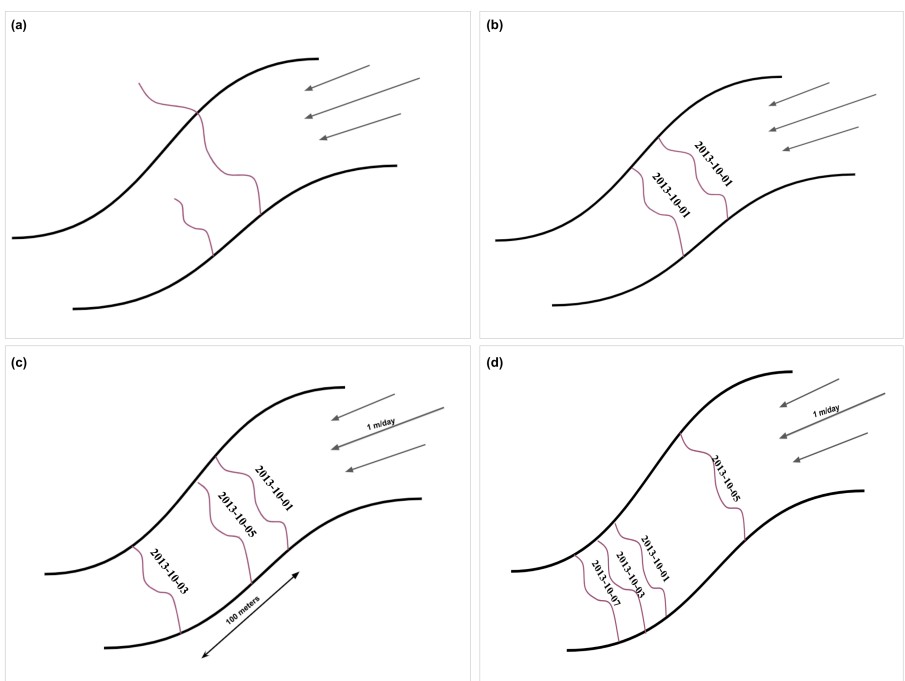

**Figure 4.** Issues associated with terminus positions from TermPicks and CALFIN: (a) incomplete terminus traces extend beyond the fjord wall; (b) duplicate traces for the same day; (c) anomalous advance where the terminus position moved faster than possible based on independent speed estimates; and (d) anomalous retreat, a similar condition to (c), where the terminus is anomalously retreated relative to the subsequent observations. Arrows show the direction of glacier flow.

are open source and therefore facilitate continuation of the terminus ablation time series as more data become available. The AERODEM is created from aerial photographs collected in the late 1970s and 1980s, with dates differing per region, and has a spatial resolution of 25 m (Korsgaard, 2021). The ArcticDEM is produced using stereo high resolution satellite images
acquired from 2013–2017, and has a spatial resolution of 2 m (Porter et al., 2022). We used the specific acquisition date of each individual DEM tile. For terminus observations prior to the acquisition date, surface elevations were estimated by linearly interpolating between the AERODEM and ArcticDEM. From the acquisition date to 2023, the ArcticDEM surface elevations were used unchanged. The potential influence of the use of these snapshot DEMs on our terminus ablation time series is further detailed in the discussion.
Although most GrIS outlet glaciers no longer have floating termini (Enderlin and Howat, 2013; Ochwat et al., 2023), observations of tabular icebergs during manual terminus delineation suggest that some termini may still reach flotation and thus ice thickness estimates cannot be estimated directly from the difference between surface and bed elevations. Since the bed elevations have the coarsest spatial resolution ($\sim$150 m), the DEMs were resampled to the BedMachine grid and we then estimated glacier thickness under the assumptions that (1) flotation occurs with an ice density of 900 kg m$^{-3}$ and seawater density of
1026 kg m$^{-3}$ and (2) the ice is grounded and the thickness is the difference between the surface and bed elevations. Where

the flotation thickness was less than the grounded ice thickness, the flotation thickness was used. Mean thickness within the terminus polygon was multiplied by the polygon area to estimate terminus volume. The terminus volume was multiplied by the density of ice (900 kg m$^{-3}$) to estimate mass within each polygon.

Consecutive terminus mass estimates were differenced to estimate the terminus mass change rate. When multiple observa-
135 tions were present within a single month, a weighted mean was used. The rates of terminus mass change were computed and weighted based on the number of days spanned between rates for that month. For months with no data, a linear interpolation based on the closest preceding and following observations was used to fill temporal gaps in the monthly time series.

## 2.2 Ice Discharge and Terminus Ablation

Ice discharge, estimated as the rate of ice flowing through a flux gate towards the terminus, was obtained from Mankoff et al.
(2021). The discharge dataset includes a larger number of glaciers per region: 71 in the NW, 16 in the CW, 13 in the SW, 88 in the SE, 52 in the CE, 8 in the NE, and 19 in the North. The frequency of discharge data from 2013–2023 increased from approximately one observation per month in 2013 to roughly three per month in 2023. Monthly averages were calculated when multiple estimates were available for a month and linear interpolation was used to fill gaps, as described for the terminus mass change time series, so that the datasets could be directly combined.

Terminus ablation was computed as the amount of ice lost from the glacier terminus over a set time period (Fig. S2),

$$A_{terminus} = D - (\Delta M / \Delta t), \tag{1}$$

where $A_{terminus}$ is terminus ablation, $D$ is ice flux towards the terminus (i.e., discharge), and $\Delta M$ is the terminus mass change rate over the time interval $\Delta t$. The only instance wherein $D - (\Delta M/\Delta t)$ would be negative is if $(\Delta M/\Delta t) > 0$ (glacier advance) and $(\Delta M/\Delta t) > D$. However, negative terminus ablation does not just mean that the terminus is advancing, it means that the
150 terminus is gaining mass at a greater rate than the glacier flow rate. Therefore, we attribute negative terminus ablation largely to interpolation effects.

Ice discharge across upstream flux gates were used as a proxy for the discharge at the terminus and do not account for potential mass loss from surface melt or lags in temporal variations in flux changes between the flux gate and the terminus. This simple approximation is justified by the common use of flux gate-based discharge estimates in studies of Greenland
dynamic mass loss (Mankoff et al., 2021; Mouginot and Rignot, 2019; King et al., 2018; Enderlin and Hamilton, 2014) and the relatively small difference in flux between gates situated 1 km and 5 km from the terminus ($\sim 5\%$) compared to the ice flux uncertainty of $\sim 10\%$ (Mankoff et al., 2021).

### 2.2.1 Analysis of Terminus Ablation Variability

Glacier seasonality was classified through analysis of the periodograms of the terminus ablation time series, in which periodic
signals were evident as peaks in spectral power density. Two primary terminus ablation periodicities were evident in the periodograms and all glaciers were assigned to one of the two corresponding classes:

**Table 1.** Uncertainties associated with various datasets.

| Dataset | Uncertainty | Source |
|---|---|---|
| Terminus positions | Pixel size × length of terminus delineation | TermPicks (Goliber et al., 2022) + CALFIN (Cheng et al., 2021) |
| Bed topography | $\sim 125 - 150$ m | BedMachine 5 (Morlighem et al., 2022) |
| Historical surface elevations | 5.4 m | AERODEM (Korsgaard, 2021) |
| Recent surface elevations | $\sim 2.3$ m | Arctic DEM (Porter et al., 2022) |
| Ice Discharge | $\sim 10\%$ | Discharge (Mankoff et al., 2021) |

(a) If the dominant period was between 10–13 months (i.e., annual) or 5.5–6.5 months (i.e., biannual), the glacier was classified as having consistent seasonal variations in terminus ablation.

(b) If the dominant period was less than 10 months but not between 5.5–6.5 months (the biannual range), the glacier was classified as erratic, with no clear seasonality and irregular spikes/dips.

Manual inspection of the terminus ablation time series also revealed multi-year changes in terminus ablation for several glaciers (Table S2). For each glacier, if the yearly average deviated by more than 20% of the the decadal average for >2 years, it was also classified as having both sub-annual variability as described above and multi-year variability in terminus ablation.

### 2.2.2 Terminus Ablation Uncertainties and Potential Bias

Terminus ablation uncertainty was calculated using standard error propagation techniques under the assumption that uncertainties in terminus position and discharge are independent. Table 1 summarizes the uncertainties associated with various datasets. Terminus location uncertainty for each delineation was assumed to be +/- 1-pixel (30 m for Landsat, 10 m for Sentinel-2) times the length of the terminus (i.e., the across-terminus width). Bed elevation uncertainty was extracted from BedMachine 5 and is typically $\sim$125–150 m. Surface elevation uncertainties are on the order of meters (Karlson et al., 2021; Korsgaard, 2021). Ice discharge uncertainties from Mankoff et al. (2021) are temporally-variable and not coincident with the terminus mass change time series, so we estimated the uncertainty in terminus ablation for each glacier using the time-averaged uncertainties in discharge and the terminus mass change rate (TableS1).

Although we estimated uncertainties in our terminus ablation rates, biases were more difficult to quantify. We attribute observations of negative terminus ablation, which indicate that mass is added to the terminus in excess of the ice flux from the interior, to underestimation of terminus mass loss caused by bias. Most negative terminus ablation estimates were obtained between February and April ($\sim$3% of total terminus ablation values), when terminus mass change rates are the most likely to be interpolated over time. During this period, terminus positions are the most difficult to identify because of low light and the presence of sea ice, and bed elevations are the most uncertain because the termini are typically the most advanced (Greene et al., 2024). Negative terminus ablation values were replaced with zeros and we recommend that all winter terminus ablation values are interpreted with caution.

## 3    Results

### 3.1    Terminus Ablation Patterns

Consistent and relatively-smooth biannual or annual variations in terminus ablation were identified for the majority of the glaciers (41 of 49). Erratic terminus ablation patterns were identified at only 8 glaciers. These annual and shorter-term variations in terminus ablation were superimposed on multi-year changes at 4 glaciers, all located in the northwestern sector of the ice sheet. Consistent biannual, consistent annual, and multi-year changes are evident in both the terminus ablation and discharge time series, indicating that they are driven by changes in ice flux rather than terminus position change. In contrast, erratic changes are only evident in the terminus ablation time series and can be attributed to large irregular calving events. Below we present three examples of terminus ablation time series to demonstrate the breadth of temporal variations in terminus ablation.

Consistent annual variations in terminus ablation are exemplified by Heimdal Glacier (Fig. 5). From 2013–2023, the glacier's terminus position was relatively steady (Fig. 5a) but small seasonal cycles of advance/retreat resulted in ∼0.4 Gt seasonal variations in mass (Fig. 5c). Overall, the glacier advanced by ∼500 meters from 2013–2023, indicating that terminus ablation was slightly less than discharge when summed over the study period. Terminus ablation remained between ∼1–2 Gt/year, with the majority of the <1 Gt/year seasonal oscillations driven by discharge variations(Fig. 5b). As the glacier advanced during 2019–2021, terminus ablation in the winter months was less than that estimated from discharge and seasonality in terminus ablation increased (Fig. 5b).

Erratic seasonal variations in terminus ablation are demonstrated with observations for Sermeq Kujalleq N2, located in the central west (Fig. 6). The glacier's terminus position was relatively steady over the decade but it seasonally retreated and advanced by  800 m (Fig 6a), driving variations in mass of ∼0.5 Gt (Fig 6c). Seasonal terminus retreat was episodic, however, resulting in highly-variable spikes in monthly terminus ablation rates that often differ from discharge by 10% or more (Fig 6b).

The clearest example of multi-year variability in terminus ablation is from Ullip Sermia, a surging glacier (Sevestre and Benn, 2015; Rignot and Kanagaratnam, 2006) (Fig. 7). The terminus slightly advanced during 2013–2017 (∼1.1 km), indicating that terminus ablation almost matched ice discharge (Fig. 7a,b), and then retreated by ∼3.2 km during 2019–2023. Seasonal variability in terminus ablation and discharge was roughly the same magnitude as the annual-mean discharge during the fast-flow period (∼1.5 Gt/yr). When discharge decreased (<0.5 Gt) at the end of 2019, discharge seasonality also decreased, but the summer terminus ablation remained high as the terminus retreated by ∼3.2 km in 4 years.

### 3.2    Regional Patterns in Terminus Ablation

Glaciers were grouped into regions following Mouginot and Rignot (2019) to identify potential regional similarities and to quantify large-scale deviations in the timing and magnitude of terminus ablation. The classification of glaciers based on their dominant frequency of terminus ablation showed notable variability between east and west Greenland (Fig. 8). On the east coast, where only 25% of the study glaciers are located, all glaciers had consistent seasonal patterns. In the southwest, one of the three glaciers had an erratic pattern, despite terminating in the same fjord. In the central west region, nine glaciers had

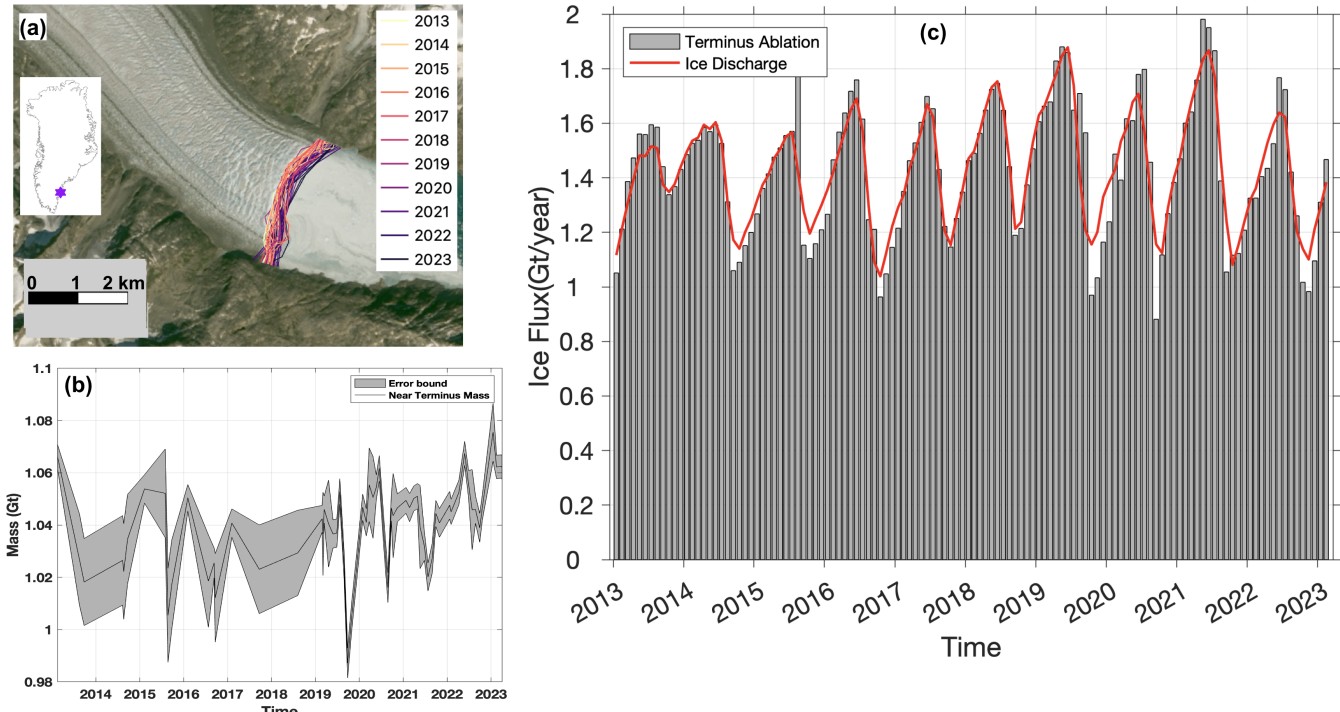

**Figure 5.** Heimdal Glacier (coordinates: -42.67188°W ,62.9001°N). (a) Terminus positions for Heimdal Glacier during 2013–2023 displayed on a true-color composite Sentinel-2 image (courtesy of European Space Agency, Copernicus). The inset of Greenland with a purple star shows the glacier location. (b) Time series of mass within the terminus polygon (Gt) with uncertainty shown as gray shading. (c) Time series for terminus ablation and discharge in gigatons per year (Gt/yr).

consistent patterns while three had erratic patterns in terminus ablation. In the northwest region, 18 glaciers had consistent seasonal patterns, whereas only four had erratic patterns in terminus ablation.

Sub-annual variations in terminus ablation rates are tens of Gt/yr greater than in discharge for all regions except the southwest (Fig. 9). For the northwest and central west, sub-annual variations in terminus ablation have a consistent seasonal pattern across the study period: terminus ablation is typically less than discharge from approximately October–March with a pronounced summer peak centered around July. Although the regional discharge is larger for the central west than the northwest (∼75 Gt/yr and ∼60 Gt/yr, respectively), the seasonal peak in terminus ablation rate is lower in the central west than the northwest

(∼100 Gt/yr and ∼110 Gt/yr, respectively). In contrast, the southwest region contains only three glaciers, each with thicknesses of less than 500 meters, resulting in terminus ablation of approximately ∼10 Gt/yr and seasonality of only ∼1 Gt/yr. Temporal variations in terminus ablation are similar across the glaciers in the central east and northeast regions, with seasonal oscillations in terminus ablation rates evident at the beginning (2013–2015) and end (2019–2022) of the time series. For the glaciers in the central east and the single glacier in the northeast, Zachariae Isstrom, there is no consistent seasonal signal during 2016–2018.

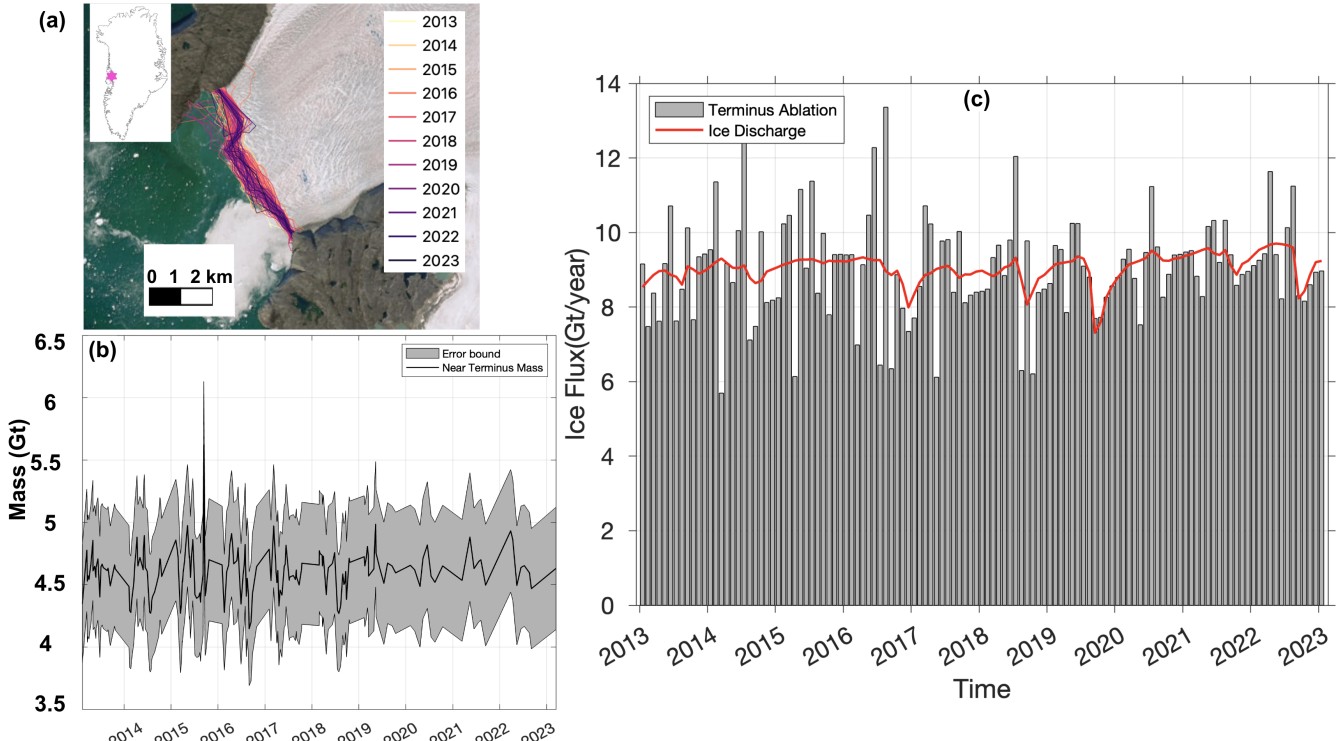

**Figure 6.** Sermeq Kujalleq N2 (coordinates: -50.61286°W ,70.3841°N). (a) Terminus positions for Sermeq Kujalleq N2 during 2013–2023 displayed on a true-color composite Sentinel-2 image (courtesy of European Space Agency, Copernicus). The inset of Greenland with a pink star shows the glacier location. (b) Time series of mass within the terminus polygon (Gt) with uncertainty shown as gray shading. (c) Time series for terminus ablation and discharge in gigatons per year (Gt/yr).

Although none of the glaciers in the southeast were identified as erratic, the regional terminus ablation rate time series is the most inconsistent, with near-monthly variations in terminus ablation rate of ∼20 Gt/yr.

To better characterize sub-annual oscillations in terminus ablation rate, regional monthly climatologies were constructed using the average monthly terminus ablation over our decade-long time series (Fig. 10a). While terminus ablation peaks in July for glaciers in the northwest, central west, and southeast, it peaks in August in the central east and northeast regions.
Interestingly, the seasonal minimum rate of terminus ablation in the southeast, southwest, and central west occurs in October/November and increases throughout the winter, whereas the seasonal minimum is not reached until March/April for the central east and northern regions. Seasonal oscillations in the regional discharge climatologies are much smaller in amplitude than terminus ablation (Fig. 10b), indicating that seasonal terminus retreat and advance are the primary drivers of seasonality in solid freshwater flux from glacier termini. However, seasonality in terminus ablation and discharge are not consistent
across study regions (Table 2). The northwest has relatively consistent discharge and terminus ablation across the seasons (∼58 Gt/yr), with the exception of a large spike in terminus ablation during the summer (∼85 Gt/yr). The central west generally has

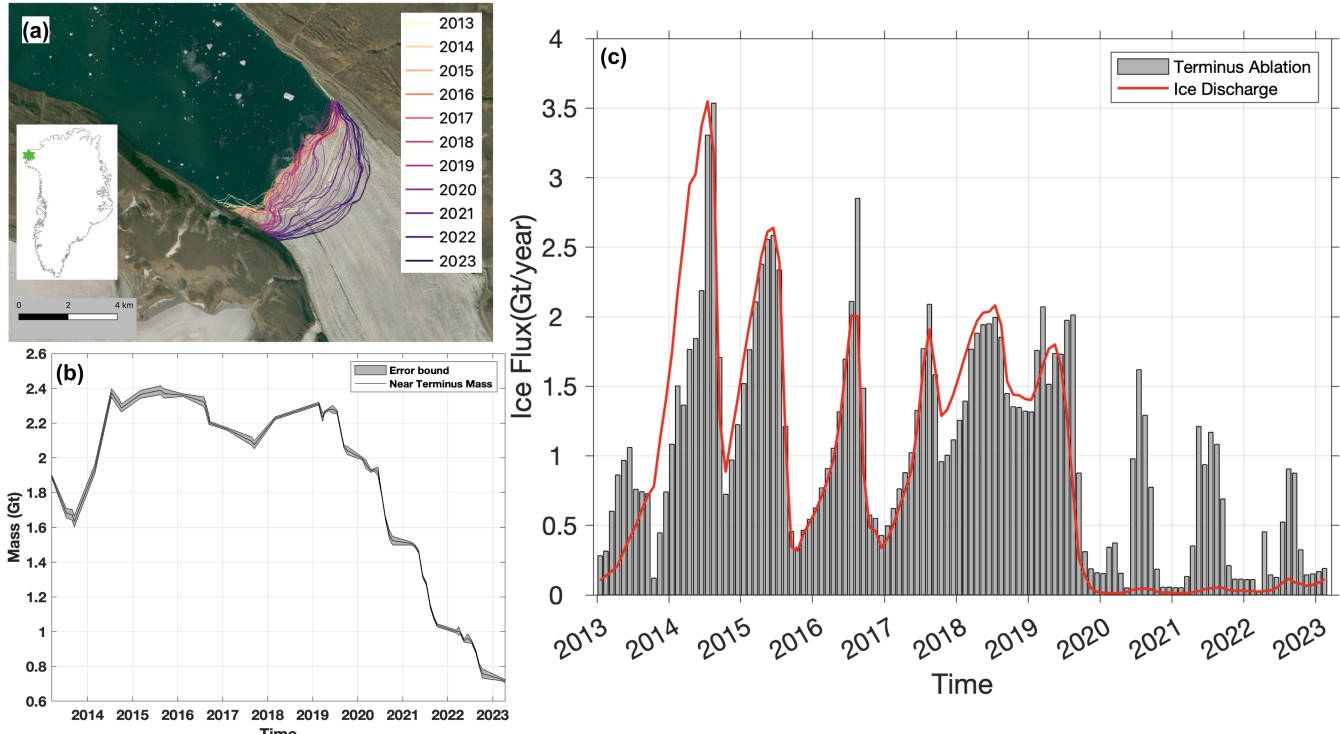

**Figure 7.** Ullip Sermia (coordinates: -67.74813°W ,76.59598°N). (a) Terminus positions for Ullip Sermia during 2013–2023 displayed on a true color composite Sentinel-2 image (courtesy of European Space Agency, Copernicus). The inset of Greenland with a green star shows the glacier location. (b) Time series of mass within the terminus polygon (Gt) with uncertainty shown as gray shading. (c) Time series for terminus ablation and discharge in gigatons per year (Gt/yr).

the highest and relatively steady values in both discharge and terminus ablation, averaging ∼72 Gt/yr other than the summer peak in terminus ablation of 85 Gt/yr. Terminus ablation in the southeast region reaches an annual minimum in November of ∼25 Gt/yr then gradually increases throughout the spring and winter (42 Gt/yr and 50 Gt/yr, respectively), reaching a seasonal peak in terminus ablation of 60 Gt/yr during the summer. For central east, discharge is stable throughout the year, but terminus ablation increases from a spring minimum of ∼30 Gt/yr to a maximum of ∼60 Gt/yr in August, with an average discharge throughout summer and fall of ∼51 Gt/yr. Seasonal patterns in discharge and terminus ablation are similar for the northeast and central west when averaged over the decade: the northeast region has relatively steady values in both discharge and terminus ablation, averaging ∼15 Gt/yr other than the summer peak in terminus ablation (24 Gt/yr). The southwest region terminus ablation remains consistently low, ranging between 7–8 Gt/yr across the year.

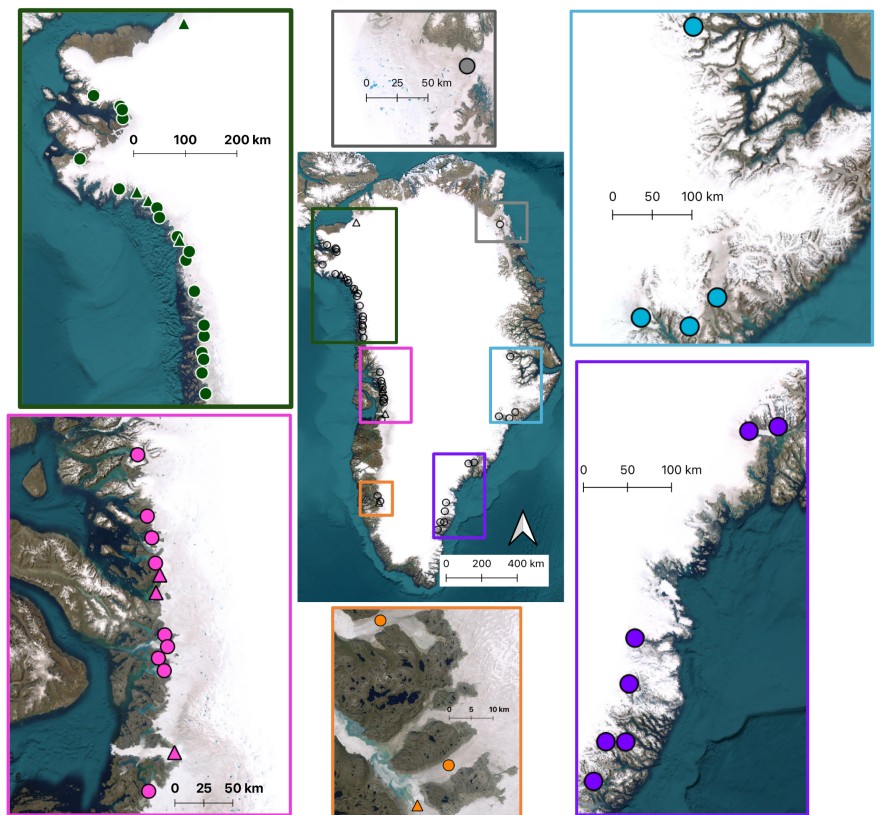

**Figure 8.** Classification of terminus ablation cycle for glaciers in different Greenland regions, colored by region. Base image is from Sentinel-2 (courtesy of European Space Agency, Copernicus). Circles indicate glaciers with consistent annual or biannual terminus ablation variations and triangles indicate glaciers with erratic terminus ablation over intra-annual timescales.

## 4 Discussion

Terminus ablation rates account for (1) mass flux changes due to changes in ice flux and 2) terminus advance and retreat, for which (2) is the key difference from traditional ice discharge estimates. For the 49 Greenland glaciers with sufficient bed elevation data to quantify mass changes due to terminus position change, we find that terminus ablation is more varied than discharge at monthly-to-seasonal time scales. This variability emphasizes the importance of accurately accounting for terminus position changes, as neglecting these changes can lead to an underestimation of seasonal fluctuations in ice fluxes from glacier termini. Terminus ablation is $\sim 7\%$ and $\sim 2\%$ lower in the spring and winter, respectively, compared to estimates based on discharge. Conversely, it is $\sim 4\%$ and $\sim 27\%$ higher in the fall and summer than discharge-based estimates (Table 2).

We also found a dichotomy in glacier behavior among the 49 glaciers studied: 41 have consistent seasonal patterns, while the remaining 8 glaciers have erratic seasonality in terminus ablation. Additionally, 4 glaciers show large interannual fluctuations, possibly driven by discharge changes or other factors such as surging. The consistent behavior may reflect more frequent,

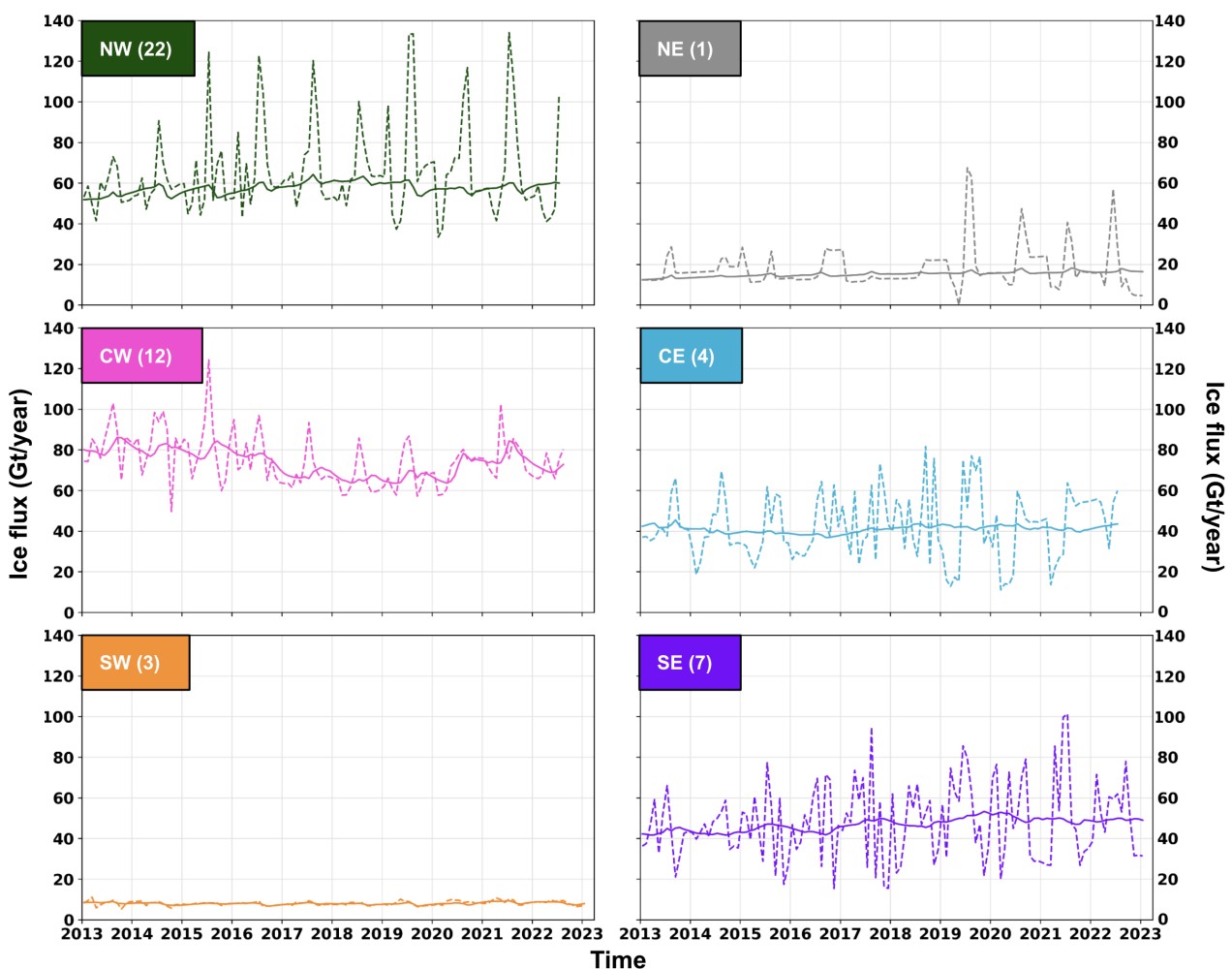

**Figure 9.** Summed terminus ablation rate (solid lines) and discharge (dashed lines) time series for each region, with colors corresponding to the regions in Figure 1. The numbers in parentheses indicate the number of study glaciers in the region.

smaller-scale calving events and/or a stronger influence of submarine melting on terminus position. We attribute the "erratic" variations in terminus ablation to irregular, large-scale calving events, which occur infrequently due to the stochastic nature of calving. However, our classification method does not always classify glaciers with such irregular events as erratic. For example, Helheim Glacier has episodic, large-scale semi-monthly calving events, yet it is not classified as erratic based on its periodogram (Fig. S3). Using our simple classification approach, we did not find regional differences in the character of terminus ablation, however, this does not necessarily imply such differences do not exist.

Terminus ablation has greater variability in both the timing and magnitude of seasonal variations compared to discharge at both individual glacier levels and across regional scales, indicating it is more sporadic and seasonally variable. This increased

**Table 2.** Terminus ablation and discharge (in Gt/yr) values averaged across seasons with one standard deviation

| Regions | Winter | | Spring | | Summer | | Fall | |
|---|---|---|---|---|---|---|---|---|
| | Discharge | Term Abl | Discharge | Term Abl | Discharge | Term Abl | Discharge | Term Abl |
| NW | 57 (2) | 59 (12) | 58 (2) | 52 (10) | 59 (3) | 85 (28) | 56 (3) | 64 (14) |
| CW | 73 (6) | 72 (9) | 71 (6) | 71 (10) | 74 (6) | 85 (12) | 76 (7) | 70 (10) |
| SW | 8 (0.5) | 8 (0.7) | 8 (0.4) | 9 (1.2) | 8 (0.4) | 9 (0.7) | 7 (0.4) | 7 (0.8) |
| SE | 47 (3) | 42 (15) | 47 (3) | 50 (15) | 47 (3) | 60 (19) | 47 (3) | 41 (19) |
| CE | 41 (2) | 39 (10) | 41 (2) | 30 (13) | 41 (2) | 50 (15) | 40 (2) | 51 (16) |
| NE | 15 (1) | 17 (6) | 15 (1) | 12 (5) | 16 (1) | 24 (16) | 15 (1) | 18 (7) |

Term Abl = Terminus Ablation; Both discharge and terminus ablation are in Gt/yr with values averaged across the entire decade based on the season (rounded to the nearest whole number). Standard deviations are provided in parentheses. Winter = December to February; spring = March to May; summer = June to August; fall = September to November.

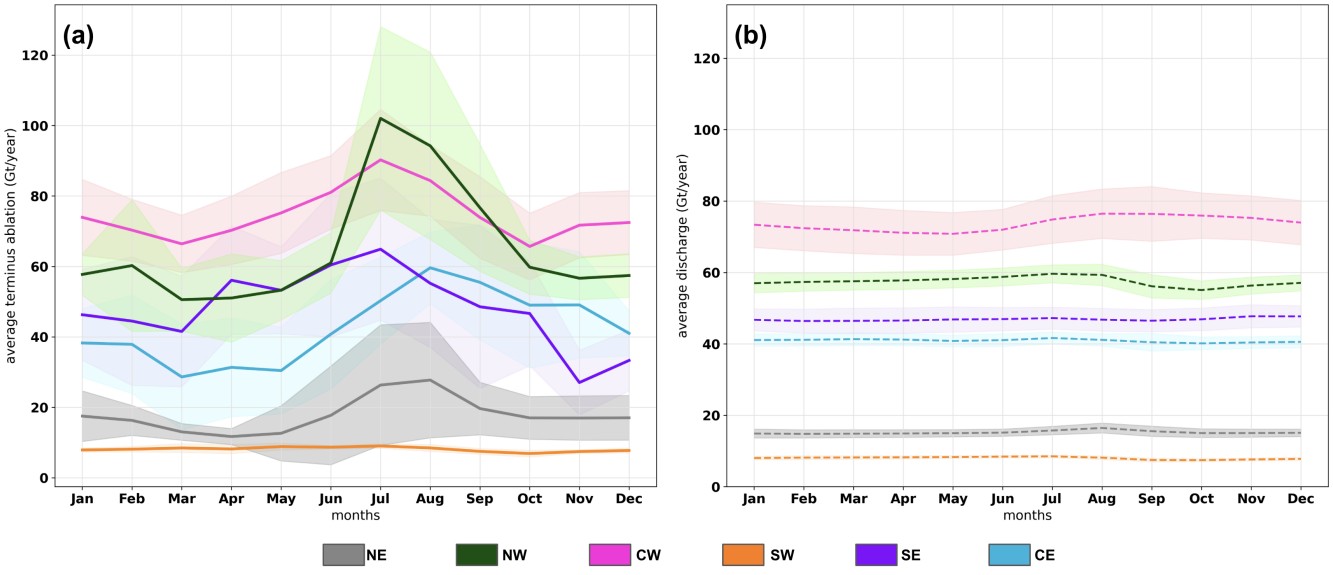

**Figure 10.** Monthly average terminus ablation rate (solid lines) and discharge (dashed lines) for 2013–2023, with colors corresponding to the regions in Fig. 1. Shading indicates the standard deviation in the monthly data.

variability is likely driven by local and regional factors. During colder months, processes such as submarine melting and calving are generally suppressed, primarily due to low air temperatures (Robel, 2017; Ekström et al., 2006), the absence of runoff that would typically drive fjord circulation (Straneo and Cenedese, 2015; Fried et al., 2018; Carroll et al., 2017), and the persistent presence of rigid ice mélange (Cassotto et al., 2015; Fried et al., 2018). Ice mélange, which is strongly associated with seasonal glacier advance (Howat et al., 2010; Moon et al., 2015), can inhibit the rotation required for slab-style calving events (Reeh

et al., 2001; Mortensen et al., 2020). As large-scale calving is most likely to occur from a terminus that is close to flotation (Amundson et al., 2010), higher terminus ablation values for glaciers like Sermeq Kujalleq (Jakobshavn) and Yngvar Nielsen Glacier (Fig. S4 and Fig. S5, respectively) can be attributed to the break-up of an ice tongue during late spring/early summer (Joughin et al., 2012; Cassotto et al., 2015; Black and Joughin, 2023). Notably, differences among glaciers, even within the same geographic region, suggest that each glacier's response is influenced by specific factors, such as geometry and terminus

type (Carr et al., 2013; Catania et al., 2020). These variations become even more apparent when comparing regional patterns. In the southwest, terminus ablation tracks the discharge cycle, indicating that smaller serac failure-style calving events remove mass at nearly the same rate as it is delivered to the terminus (Robel, 2017; Fahrner et al., 2021). For the central west, previous studies have shown that both seasonality in ice mélange extent and submarine melting can influence seasonal terminus position change (Moon and Joughin, 2008; Mortensen et al., 2020; Howat et al., 2010). However, their relative importance varies with

glacier geometry, driving widespread terminus retreat (Greene et al., 2024; Fried et al., 2018) and enhanced terminus ablation throughout the first half of the melt season that peaks in July (Rignot et al., 2016; Carroll et al., 2015, 2016). As with the central west sector, seasonal retreat of glaciers in the northwest has been in part attributed to the break-up of sea ice and ice mélange (Carr et al., 2013), and our terminus ablation time series indicate that the near-coincident retreat of glaciers after an early-summer maximum extent causes a large seasonal spike in terminus ablation (Greene et al., 2024)(Fig. 10).

Several biases and uncertainties influence the accuracy of terminus ablation rate estimates, including data availability, interpolation artifacts, and potential errors in datasets. The pervasive fall/winter drop in terminus ablation rate to below discharge rate (Table 2, Fig. 10) is likely due to a combination of decreased rates of iceberg calving and submarine melting, as well as data availability and quality. Terminus position observations are sparse in the winter months and the terminus position can be difficult to identify in the late winter/early spring when ice mélange is densely packed within fjords. We attribute unphysical

negative terminus ablation rates primarily to interpolation artifacts in the winter months, when the $\sim 3\%$ of the negative terminus ablation estimates occur (Fig. S6). Our assumption that mass change between the terminus and the flux gate due to surface accumulation and ablation is minimal may slightly bias terminus ablation estimates, but it is unlikely to cause negative terminus ablation rates because the close proximity of the flux gate and terminus and fast ice flow minimizes the travel time to the terminus. Since terminus position varies on seasonal to decadal time scales, surface mass balance changes between the flux gate

and terminus could potentially bias terminus ablation estimates. However, seasonal terminus retreat is relatively small ($\sim 1$ km) and only 7 glaciers retreated >3 km from 2013-2023 (Fig. S10 and S11). For the glaciers with the largest magnitude of retreat, surface speeds are $\sim 2$-6 km per year (Gardner et al., 2019) such that the maximum surface mass balance correction between the flux gate and the most extended terminus position would only be on the order of a few meters of ice thickness. Given that ice thickness at the flux gates generally exceeds several hundred meters, the surface mass balance adjustment would be «10% of

the total ice thickness throughout the entire study period. An independent estimate of surface mass balance between flux gates and termini for 213 of Greenland's outlet glaciers suggests it contributes only $\sim 3\%$ to decadal terminus ablation (Kochtitzky et al., 2023). When averaged over the 2013–2023 study period, our terminus ablation estimates are in good agreement with the decadal-scale estimates of (Kochtitzky et al., 2023) (Fig. S9). Differences between these terminus ablation estimates are likely due to variations in the temporal resolution of the terminus position and discharge time series (e.g., monthly vs. decadal), the

source of discharge data (Mankoff et al., 2021; Kochtitzky and Copland, 2022), and the treatment of ice thickness (single value vs. dynamic) (Fahrner et al., 2025). These differences exceed the adjustments made for surface mass balance. Further support for the relatively small mass change caused of surface accumulation and ablation between the flux gates and termini comes from Greenland's peripheral glaciers, which are generally slower-flowing than outlet glaciers; for peripheral glacier, surface mass balance adjustments decreased by 0.2 Gt/yr from 1999–2018 due to terminus retreat and were only $\sim 10\%$ of discharge

on average (Bollen et al., 2022). Therefore, we conclude that surface mass change between the terminus and the flux gate is within the overall uncertainty range.

Biases in bed elevation near glacier termini that are in excess of the quoted uncertainties in BedMachine may influence our terminus ablation estimates but cannot be constrained with observations. Biases in bed elevation are particularly important for termini near flotation because they could cause floating ice to be falsely mapped as grounded, resulting in ice thickness over-

320 estimation. The identification of floating ice is also dependent on the accuracy of density estimates. Here we estimate terminus mass using a slightly smaller density (900 kg m$^{-3}$) than used to calculate discharge (917 kg m$^{-3}$) by Mankoff et al. (2021) in order to account for the influence of crevassing on near-terminus ice density. Although the differences in density between datasets (<2%) should have a relatively small influence on terminus ablation estimates, they can influence the identification of floating ice. The amplitude of seasonal oscillations in terminus ablation rate will be exaggerated where floating tongues

seasonally form and disintegrate, but most glaciers in Greenland no longer maintain perennial ice tongues (Enderlin and Howat, 2013; Catania et al., 2020) and we expect seasonal floating tongues to be fairly short. Given uncertainties in the extent of floating tongues, intercomparisons among GRACE-derived mass loss (e.g., Velicogna et al. (2020); Sasgen et al. (2020)), altimetry (Shepherd et al., 2019; Felikson et al., 2017), and input-output methods (Mouginot et al., 2019; Colgan et al., 2019) should continue to use discharge, which reflects the mass flux across the gate. For grounded termini, the impact on sea-level

is governed by the fraction of ice above flotation, since ice already displacing water does not contribute to sea-level rise upon calving. In such contexts, incorporating total ablation may overestimate the sea-level relevant mass loss unless the floating fraction is taken into account. For analysis of seasonal processes at the fjord scale, however, the terminus ablation time series are more appropriate even when considering their uncertainties.

While the data suggest that terminus change and discharge influence each other, we cannot draw definitive conclusions

about causality from these observations because of the bidirectional coupling between terminus change and discharge (King et al., 2020; Dryak and Enderlin, 2020). However, our analysis indicates that terminus position changes strongly influence the seasonal timing and magnitude of ice fluxes from Greenland's outlet glaciers. Iceberg calving influences the structural properties of ice mélange and mixing of water masses within fjords (Amundson et al., 2010; Bassis and Jacobs, 2013; Robel, 2017), which in turn influence terminus stability. Different calving styles influence glacier mass balance in distinct ways, with

some glaciers experiencing episodic large-scale calving events while others exhibit more gradual, continuous retreat (Cassotto et al., 2015; Catania et al., 2020). While previous analyses of terminus position at seasonal time scales have largely focused on the relationship between terminus change and glacier speed (King et al., 2020; Black and Joughin, 2022, 2023; Moon and Joughin, 2008), we focus on the direct influence of terminus position change on mass loss. Using terminus position data from previous studies, supplemented with more recent manual delineations created by following the approach recommended by

Goliber et al. (2022), we highlight the contribution of seasonal terminus retreat to ice mass loss.For example, in the central west, seasonal terminus position changes ranging from 1500 to 2500 m result in an estimated mass loss of approximately 80–100 Gt/year (Fig. 10, S7 and S8). The timing and magnitude of iceberg calving can also control when and where iceberg meltwater enters the surrounding fjords and ocean basins, which can strongly influence the fate of the freshwater (Luo et al., 2016; Moon et al., 2018). Although our monthly terminus ablation record only includes 49 outlet glaciers, it shows that failure to account for terminus position change typically results in overestimates of fall/winter ice fluxes and underestimates of spring/summer ice fluxes by tens of Gt/yr. Depending on the glacier, terminus ablation rates can have consistent sub-annual or annual oscillations, erratic spikes and dips, and/or multi-year changes that are either superimposed on sub-annual variability or can drive changes in the sub-annual variability in terminus ablation rates. Even though terminus position change is a relatively small component of Greenland mass loss over decadal time scales (Kochtitzky and Copland, 2022; Greene et al., 2024), it is the primary driver of monthly-to-seasonal variations in terminus ablation. Seasonal precision in the estimates of terminus ablation is particularly important for understanding freshwater flux impacts on downstream ecosystems, fjord productivity, and ocean circulation. Thus, future estimates of Greenland freshwater fluxes should account for these seasonal terminus dynamics to capture their influence on sensitive environmental systems.

## 5 Conclusions

Our study provides a comprehensive, space-time-resolved understanding of terminus ablation, which is essential for accurately assessing mass loss from Greenland's marine-terminating glaciers. Using publicly-available data on terminus position, glacier thickness, and ice velocity, we characterized the sub-annual variability in terminus ablation for 49 marine terminating glaciers between from 2013–2023. Our analysis shows that terminus ablation—driven by both terminus retreat/advance and changes in ice flux—introduces considerable variability in mass loss that is not captured by discharge measurements alone. This discrepancy, which amounts to an estimated ∼14 Gt/year, emphasizes the critical role of terminus position changes for avoiding underestimation of seasonal fluctuations in glacier ice fluxes.

Potential underestimation of mass loss has critical implications for understanding freshwater fluxes into Greenland's fjords and surrounding ocean basins. The seasonal divergence underlines the importance of considering terminus position changes to avoid underestimating ice loss in warmer months and overestimating it during colder months. Therefore, future studies aiming to model ice-ocean interactions and freshwater fluxes to the ocean should prioritize high-resolution terminus ablation data to capture the full extent of seasonal and sub-annual variability. By examining terminus ablation data in more detail, more-accurate rates of localized ice loss can be created to better understand changes in regional and large-scale ocean dynamics.

*Code availability.* The code used for the terminus ablation is available as a public GitHub repository (https://doi.org/10.5281/zenodo.14042739) (KC et al., 2024a).

*Data availability.* Dataset for the paper is available through the Arctic Data Center (https://doi.org/10.18739/A2JW86P8F) (KC et al., 2024b).

*Author contributions.* Conceptualization: Aman KC, Ellyn Enderlin, Dominik Fahrner, Twila Moon, and Dustin Carroll; Data curation: Aman KC, Ellyn Enderlin, and Dominik Fahrner; Formal analysis: Aman KC and Ellyn Enderlin; Funding acquisition: Ellyn Enderlin, Twila Moon, and Dustin Carroll; Methodology: Aman KC, Ellyn Enderlin, and Dominik Fahrner; writing — original draft: Aman KC; writing — review and editing: Ellyn Enderlin, Dominik Fahrner, Twila Moon, and Dustin Carroll; Project Administration: Ellyn Enderlin, Twila Moon, and Dustin Carroll

*Competing interests.* The authors declare no competing interests.

*Acknowledgements.* We acknowledge support from members of CryoGARS-Glaciology group who helped in digitizing terminus traces. The work is supported by the National Science Foundation under Arctic Natural Sciences Grant 2052561, 2052551, and 2052549. Arctic DEMs were provided by the Polar Geospatial Center under NSF-OPP awards 1043681, 1559691, and 1542736. We are grateful to constructive reviews from Chad Greene (NASA Jet Propulsion Laboratory) and one anonymous reviewer that improved the quality of this manuscript.

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
