# Peer review of "Seasonality in Terminus Ablation Rates for the Glaciers in Greenland (Kalaallit Nunaat)"

_EGUsphere, 2024_

## Referee Comment (RC1)

**Seasonality in Terminus Ablation Rates for the Glaciers in Kalaallit Nunaat (Greenland)**
Manuscript by Aman KC et al.

*Reviewed by Chad Greene of NASA Jet Propulsion Laboratory,*
*December 18, 2024.*

**Overall impression**

In this paper, Aman KC and colleagues describe the seasonal advance and retreat of 49 marine-terminating glaciers around Greenland. The paper is well written, and the language and figures are polished to perfection. The technical approach employed here is strong and the methods are well described. I know from experience that the task of combining multiple terminus datasets is not an easy one, yet the authors have done an excellent job of making it *seem* easy, due to the well-reasoned logic they employ and clarity with which they describe it. Overall, the manuscript is of high quality and I believe it will be ready for publication if the results can be framed with a little more context.

The greatest room for improvement lies in the presentation of the current results in relation to previous studies that have conducted similar types of analysis. **I would like to see clear and direct statements in the abstract and discussion/conclusion sections that offer insight into how the present results confirm, upend, or reshape our understanding of previous studies.** Many previous studies have presented observations of terminus advance and retreat in Greenland, so what makes this one different? How do the current results change the narrative, what's different about this study, what's the big takeaway, and what guidance might the authors give to IMBIE, the IPCC, ice sheet modelers, or whomever might want to build on this work? Do these results support/strengthen our understanding of something that was already known, or introduce something new?

Although the discussion section includes a review of previous studies of physical processes that can cause seasonal advance and retreat, what's missing is a direct comparison of the present results to previous results. Several previous assessments of terminus variability are even cited in this paper, but they are constrained to the Introduction section as a means of providing motivation for the present study. I'd like to see the authors circle back at the end to place the new results in context of previous studies by Black, Joughin, Moon, Kochtitzky, Catania, Fried, Wood, Schild, Cassotto, Howat, Greene, etc. I don't mean to be too prescriptive here, so adjust that list according to taste. Direct quantitative comparisons to all those previous papers may be impossible due to differences in observation periods, but the authors have the intuition and expertise here to guide readers to find key similarities or difference between the present study and previous ones.

*Note: The data and code availability statements in this paper point to links that do not currently exist. Accordingly, this review does not apply to the data or code, and I'm taking it on blind faith that the data is of adequate quality and will be made available as promised.*

**Minor comments**

**Line 11** Regarding this sentence in the abstract:

*"for the northwest and central west sectors, where the fraction of outlet glaciers included in our estimates is greatest, the average difference between the annual maximum and minimum in terminus ablation are ~51Gt/yr and ~25Gt/yr, respectively, compared to only ~5Gt/yr for discharge."*

That's really hard to parse, especially for someone who has not yet read the paper. I recommend rewording to reduce the number of comma-bound clauses, because they tend to break the flow and force the reader to mentally keep track of too many little concepts and their relationships to each other.

The sentence also requires the reader to have a prior understanding of what is meant by the somewhat ambiguous phrase "annual maximum and minimum terminus ablation", which I can try to guess the meaning of, but not confidently enough to understand the significance of its quantified rates of 51 Gt/yr vs 25 Gt/yr.

**Line 115 and elsewhere** I think AERODEM should be written in all caps?
https://www.nodc.noaa.gov/archive/arc0088/0145405/1.1/data/0-data/G150AERODEM/

**Figure 4** is very nice. I appreciate how the simplicity of the diagram focuses attention on the issues that can cause anomalous terminus position picks, so the figure will be useful to anyone who has not encountered these issues firsthand.

**Line 157-168** The equations in Sec 2.2.1 are relatively innocuous, but I think they're relatively standard mathematical formulations, right? If there's something nonstandard going on here, be sure to mention it explicitly, otherwise I think it would be sufficient to say you plot the power spectral density of the terminus ablation time series and remove the equations.

Rather than reading equations, **in this section I would like to see PSD plot(s) that illustrate seasonal vs erratic glaciers because a major conclusion of the paper depends entirely on how the PSD is interpreted.** Showing those PSD plots will help readers gain an intuition for how the results are obtained and how sensitive the overall findings might be to subjective differences in interpretation.

**Figures 5-7** I think I'm missing something here, because panel c is presented at monthly resolution, whereas the observations in panel b are presented in irregular intervals. I would expect the overall Mass time series to be the integral of the Ice Flux time series, but that's not what these panels look like. Please explain how they're related to each other.

---

## Author Comment (AC1)

We thank the reviewer for the thoughtful comments and suggestions for the manuscript. Below, we outline our changes for revision based on the comments. The reviewer comments are shown in **blue** and our responses are shown in **black** for clarity.

In this paper, Aman KC and colleagues describe the seasonal advance and retreat of 49 marine terminating glaciers around Greenland. The paper is well written, and the language and figures are polished to perfection. The technical approach employed here is strong and the methods are well described. I know from experience that the task of combining multiple terminus datasets is not an easy one, yet the authors have done an excellent job of making it seem easy, due to the well-reasoned logic they employ and clarity with which they describe it. Overall, the manuscript is of high quality and I believe it will be ready for publication if the results can be framed with a little more context.

The greatest room for improvement lies in the presentation of the current results in relation to previous studies that have conducted similar types of analysis. I would like to see clear and direct statements in the abstract and discussion/conclusion sections that offer insight into how the present results confirm, upends, or reshape our understanding of previous studies. Many previous studies have presented observations of terminus advance and retreat in Greenland, so what makes this one different? How do the current results change the narrative, what's different about this study, what's the big takeaway, and what guidance might the authors give to IMBIE, the IPCC, ice sheet modelers, or whomever might want to build on this work? Do these results support/strengthen our understanding of something that was already known, or introduce something new?

Although the discussion section includes a review of previous studies of physical processes that can cause seasonal advance and retreat, what's missing is a direct comparison of the present results to previous results. Several previous assessments of terminus variability are even cited in this paper, but they are constrained to the Introduction section as a means of providing motivation for the present study. I'd like to see the authors circle back at the end to place the new results in context of previous studies by Black, Joughin, Moon, Kochtitzky, Catania, Fried, Wood, Schild, Cassotto, Howat, Greene, etc. I don't mean to be too prescriptive here, so adjust that list according to taste. Direct quantitative comparisons to all those previous papers may be impossible due to differences in observation periods, but the authors have the intuition and expertise here to guide readers to find key similarities or difference between the present study and previous ones.

Thank you for pointing out that the big take-aways from our analysis are not necessarily clear as is. The main objective of our research is to demonstrate that  dynamic mass loss rates on seasonal scales are strongly influenced by terminus position change even if direct mass loss from terminus retreat is small over decadal time scales. We did not include a discussion of the terminus position change record in the original submission because the focus of the study is on mass loss whereas the majority of the previous studies of terminus position focused on length or area, not mass, so an intercomparison is somewhat difficult. However, based on the reviewer's

suggestion, we plan to include an additional figure that highlights seasonal terminus position changes at the study glaciers and the following text in the discussion on line 305:

…which in turn influence terminus stability. *"Different calving styles influence glacier mass balance in distinct ways, with some glaciers experiencing episodic large-scale calving events while others exhibit more gradual, continuous retreat (Cassotto et al., 2019; Catania et al., 2020). While previous analyses of terminus position at seasonal time scales have largely focused on the relationship between terminus change and glacier speed (King et al., 2020; Black et al., 2022, 2023; Moon et al., 2008), we focus on the direct influence of terminus position change on mass loss. Using terminus position data from a number of previous studies, supplemented by more recent manual delineations created following the approach recommended by Goliber et al. (2022), we highlight the contribution of seasonal terminus retreat to ice mass loss. For example, as illustrated in the Fig. R1 and R2 and, Fig. 10, these seasonal terminus position changes on the order of 2500 m to 1500 m are associated with approximately ~80-100 Gt of mass loss per year in the central west."*

We will also include the following plots for each region in the supplement, that shows the seasonal retreat as a direct contributor to mass loss.

[Figure]

[Figure]

**Fig R1: Terminus position time series for each region, with colors corresponding to the regions in Figure 1.** Each line indicates an individual glacier corresponding to the names in the lower panel highlighted by the respective region. The numbers in parentheses indicate the number of study glaciers in the region.

[Figure]

[Figure]

**Fig R2: Seasonality of terminus position time series for each region, with colors corresponding to the regions in Figure 1.** Each line indicates an individual glacier corresponding to the names in the lower panel highlighted by the respective region. The numbers in parentheses indicate the number of study glaciers in the region.

We will add the following text in the discussion on line 387, where we compare our glacier-specific terminus ablation estimates with the decadal-scale values reported by Kochtitzky et al. (2023) for glaciers included in both studies.

…in the northwest (Fig. 10). *"Greenland wide decadal terminus ablation analysis was performed by Kochtitzky et al. (2023).  On comparing those estimates for the glaciers included in both studies, we found most of our estimates were in general agreement with the values reported in Kochtitzky et al. (2023). The differences could be a result of varying temporal resolution (monthly vs decadal), discharge data source (Mankoff et al. 2020 and  Kochtitzky et al. 2023), and frequency of terminus delineations."*

**References:**
Black, T. E. and Joughin, I.: Multi-decadal retreat of marine-terminating outlet glaciers in northwest and central-west Greenland, Cryosphere, 16, 807–824, https://doi.org/10.5194/TC-16-807-2022, 2022.

Black, T. E. and Joughin, I.: Weekly to monthly terminus variability of Greenland's marine-terminating outlet glaciers, Cryosphere, 17, 1–13, https://doi.org/10.5194/TC-17-1-2023, 2023.

Catania, G. A., Stearns, L. A., Moon, T. A., Enderlin, E. M., and Jackson, R. H.: Future Evolution of Greenland's Marine-Terminating Outlet Glaciers, Journal of Geophysical Research: Earth Surface, 125, https://doi.org/10.1029/2018JF004873, 2020.

Cassotto, R., Fahnestock, M., Amundson, J. M., Truffer, M., Boettcher, M. S., De La Peña, S., & Howat, I. (2019). Non-linear glacier response to calving events, Jakobshavn Isbræ, Greenland. Journal of Glaciology, 65(249), 39–54. doi:10.1017/jog.2018.90

King, M. D., Howat, I. M., Candela, S. G., Noh, M. J., Jeong, S., Noël, B. P., van den Broeke, M. R., Wouters, B., and Negrete, A.: Dynamic ice loss from the Greenland Ice Sheet driven by sustained glacier retreat, Communications Earth & Environment 2020 1:1, 1, 1–7, https://doi.org/10.1038/s43247-020-0001-2, 2020

Kochtitzky, W., Copland, L., King, M., Hugonnet, R., Jiskoot, H., Morlighem, M., Millan, R., Khan, S. A., and Noël, B.: Closing Greenland's Mass Balance: Frontal Ablation of Every Greenlandic Glacier From 2000 to 2020, Geophysical Research Letters, 50, e2023GL104 095, https://doi.org/10.1029/2023GL104095, 2023

Mankoff, K. D., Fettweis, X., Langen, P. L., Stendel, M., Kjeldsen, K. K., Karlsson, N. B., Noël, B., Van Den Broeke, M. R., Solgaard, A., Colgan, W., Box, J. E., Simonsen, S. B., King, M. D., Ahlstrøm, A. P., Andersen, S. B., and Fausto, R. S.: Greenland ice sheet mass balance from 1840 through next week, Earth System Science Data, 13, 5001–5025, https://doi.org/10.5194/ESSD-13-5001-2021, 2021.

Moon, T. and Joughin, I.: Changes in ice front position on Greenland's outlet glaciers from 1992 to 2007, Journal of Geophysical Research: Earth Surface, 113, 2022, https://doi.org/10.1029/2007JF000927, 2008

Note: The data and code availability statements in this paper point to links that do not currently exist. Accordingly, this review does not apply to the data or code, and I'm taking it on blind faith that the data is of adequate quality and will be made available as promised.

The correct links have been updated.

Minor comments:
Line 11 Regarding this sentence in the abstract:
*"for the northwest and central west sectors, where the fraction of outlet glaciers included in our estimates is greatest, the average difference between the annual maximum and minimum in terminus ablation are ~51Gt/yr and ~25Gt/yr, respectively, compared to only ~5Gt/yr for discharge."*

That's really hard to parse, especially for someone who has not yet read the paper. I recommend rewording to reduce the number of comma-bound clauses, because they tend to break the flow and force the reader to mentally keep track of too many little concepts and their relationships to each other.

The sentence also requires the reader to have a prior understanding of what is meant by the somewhat ambiguous phrase "annual maximum and minimum terminus ablation", which I can try to guess the meaning of, but not confidently enough to understand the significance of its quantified rates of 51 Gt/yr vs 25 Gt/yr.

Thank you for the helpful feedback. We will reword the sentence for clarity as:
*"At regional scales, seasonal variations in terminus ablation are much larger than those in discharge. We focus our intercomparison of discharge and terminus ablation on the northwest and central west sectors, where the highest fraction of outlet glaciers is included in our terminus ablation dataset. For these sectors, terminus ablation varies by approximately 51 Gt/yr and 25 Gt/yr, respectively, over each year. In contrast, the corresponding variation in discharge is only ~5 Gt/yr."*

Line 115 and elsewhere I think AERODEM should be written in all caps?
https://www.nodc.noaa.gov/archive/arc0088/0145405/1.1/data/0-data/G150AERODEM/

Thank you for the suggestion. We have changed "AeroDEM" to "AERODEM".

Figure 4 is very nice. I appreciate how the simplicity of the diagram focuses attention on the issues that can cause anomalous terminus position picks, so the figure will be useful to anyone who has not encountered these issues firsthand.

We are glad that you found the simplicity of Figure 4 effective in highlighting the key issues.

**Line 157-168** The equations in Sec 2.2.1 are relatively innocuous, but I think they're relatively standard mathematical formulations, right? If there's something nonstandard going on here, be sure to mention it explicitly, otherwise I think it would be sufficient to say you plot the power spectral density of the terminus ablation time series and remove the equations. Rather than reading equations, in this section I would like to see PSD plot(s) that illustrate seasonal vs erratic glaciers because a major conclusion of the paper depends entirely on how the PSD is interpreted. Showing those PSD plots will help readers gain an intuition for how the results are obtained and how sensitive the overall findings might be to subjective differences in interpretation.

We agree that the equations presented are standard mathematical formulations for calculating the power spectral density of the terminus ablation time series and we will remove them. Regarding your suggestion to include PSD plots, we initially explored this approach. However, the periodogram values exhibit considerable variability across different glaciers, making it challenging to present a clear visual distinction between seasonal and erratic behaviors through plots alone (shown below).

[Figure]

This variability can obscure meaningful patterns and increase the likelihood of subjective interpretation. Given these challenges, we chose to present the classification in tabular form, as it more concisely and objectively conveys the primary results without relying on interpretative nuances from the PSD plots.

We will include the following table in the supplement after formatting it to meet the journal's requirements:

| | glacier | region | frequency | lat | lon | seasonality | long-term interannuual |
|---|---|---|---|---|---|---|---|
| 0 | Sermeq_Avannarleq | CW | 12.100000 | 70.0853 | -50.2468 | C | N |
| 1 | Innaqqissorsuup_Oqquani_Sermeq | NW | 4.518519 | 76.3833 | -62.7665 | E | N |
| 2 | Sermeq_Avanarleq | NW | 12.100000 | 73.9410 | -55.7679 | C | N |
| 3 | Helheim_Gletsjer | SE | 12.100000 | 66.3735 | -38.3067 | C | N |
| 4 | Tuttulipaluup_Sermia | NW | 12.100000 | 77.7000 | -66.2330 | C | N |
| 5 | Kangerlussuup_Sermia | CW | 12.100000 | 71.4588 | -51.3101 | C | N |
| 6 | Kangerlussuaq_Gletsjer | CE | 12.200000 | 68.6746 | -33.0760 | C | N |
| 7 | Qaqujaarsuup_Sermia | NW | 5.761905 | 77.5167 | -65.6664 | C | N |
| 8 | Apuseerajik | SE | 12.100000 | 66.3820 | -37.5439 | C | N |
| 9 | Nordenskiold_Gletsjer | NW | 60.500000 | 75.8248 | -59.0399 | C | Y |
| 10 | Zachariae_Isstrom | NE | 11.000000 | 78.9333 | -21.0000 | C | N |
| 11 | Sermeq_Kujalleq_N1 | CW | 12.200000 | 69.9960 | -50.1596 | C | N |
| 12 | Sermeq_Silarleq | CW | 12.200000 | 70.8268 | -50.7627 | C | N |
| 13 | Sermersuaq | NW | 3.270270 | 79.4462 | -63.3957 | E | N |
| 14 | Tuttulikassaap_Sermia | NW | 12.100000 | 74.9618 | -57.0355 | C | N |
| 15 | Nansen_Gletsjer | NW | 6.050000 | 75.7759 | -58.8283 | E | Y |
| 16 | Sermeq_Kujalleq_N3 | NW | 12.100000 | 73.8317 | -55.5825 | C | N |
| 17 | Sermilik | CW | 12.100000 | 70.6333 | -50.6167 | C | N |
| 18 | Ikissuup_Sermersua | NW | 12.777778 | 74.2307 | -55.8275 | C | N |
| 19 | Christian_IV_Gletsjer | CE | 12.200000 | 68.7000 | -30.6167 | C | N |
| 20 | Qeqertaarsuusarsuup_Sermia | NW | 12.200000 | 77.6564 | -65.9679 | C | N |
| 21 | Sermeq_Avannarleq_N | CW | 4.461538 | 70.5454 | -50.4896 | E | N |
| 22 | Kangilernata_Sermia | CW | 12.100000 | 69.9009 | -50.3420 | C | N |
| 23 | Qeqertarsuup_Sermia | NW | 6.052632 | 73.5932 | -55.5306 | C | N |
| 24 | Rimfaxe | SE | 12.200000 | 63.3050 | -42.3669 | C | N |
| 25 | Kangerluarsuup_Sermia | NW | 12.600000 | 77.6934 | -68.5829 | C | N |
| 26 | Narsap_Sermia | SW | 12.100000 | 64.6672 | -49.8576 | C | N |
| 27 | Perlerfiup_Sermia | CW | 12.888889 | 70.9909 | -50.9227 | C | N |
| 28 | Bernstorff | SE | 12.100000 | 63.8846 | -41.7654 | C | N |
| 29 | Saqqarliup_Sermia | CW | 11.000000 | 68.8667 | -50.2833 | C | N |

| | | | | | | | |
|---|---|---|---|---|---|---|---|
| 30 | Akullersuup_Sermia | SW | 12.200000 | 64.3833 | -49.4779 | C | N |
| 31 | Apusiigajik | SE | 12.200000 | 63.2926 | -41.9168 | C | N |
| 32 | Qeqertat_Timaanni_Sermeq | NW | 8.642857 | 76.3000 | -61.7666 | E | N |
| 33 | Yngvar_Nielsen_Gletsjer | NW | 11.000000 | 76.3334 | -64.0831 | C | N |
| 34 | Ullip_Sermia | NW | 12.200000 | 76.5797 | -67.6260 | C | N |
| 35 | Sermeq_Kujalleq | CW | 5.809524 | 69.1833 | -49.8000 | E | Y |
| 36 | Naajarsuit_Sermiat | NW | 12.100000 | 73.2500 | -55.0833 | C | N |
| 37 | Heimdal | SE | 12.200000 | 62.9052 | -42.6851 | C | N |
| 38 | Graulv | SE | 12.100000 | 64.3500 | -41.5667 | C | N |
| 39 | Frederiksborg_Gletsjer | CE | 11.600000 | 68.4557 | -31.6620 | C | N |
| 40 | Illullip_Sermia | NW | 12.100000 | 74.4167 | -55.9666 | C | N |
| 41 | Rink_Gletsjer_N | NW | 5.761905 | 76.2167 | -60.9999 | C | N |
| 42 | Issuusarsuit_Sermiat | NW | 12.200000 | 76.0667 | -60.6333 | C | Y |
| 43 | Daugaard_Jensen_Gletsjer | CE | 11.500000 | 71.7500 | -29.0000 | C | N |
| 44 | Sermeq_Kujalleq_N2 | CW | 2.520833 | 70.4054 | -50.5364 | E | N |
| 45 | Dietrichson_Gletsjer | NW | 12.300000 | 75.4582 | -58.0637 | C | N |
| 46 | Kangiata_Nunaata_Sermia | SW | 4.066667 | 64.2966 | -49.6102 | E | N |
| 47 | Sverdrup_Gletsjer | NW | 11.600000 | 75.6195 | -57.9704 | C | N |
| 48 | Eqip_Sermia | CW | 12.200000 | 69.8080 | -50.1851 | C | N |

**Figures 5-7** I think I'm missing something here, because panel c is presented at monthly resolution, whereas the observations in panel b are presented in irregular intervals. I would expect the overall Mass time series to be the integral of the Ice Flux time series, but that's not what these panels look like. Please explain how they're related to each other.

In Figures 5-7, the relationship between panels b and c lies in how terminus mass change and ice flux are derived and processed.
Panel c presents ice flux components in a standardized monthly time series. Because these mass fluxes are from different datasets with varying temporal resolutions, they are either linearly interpolated or weighted to ensure comparability at a monthly scale.

Panel b represents near-terminus mass changes, which are based on observed terminus position data and are often irregular in time. This panel highlights the uncertainty in total mass estimates, which are primarily from terminus position records. The terminus polygons used for mass estimation are constructed using each available terminus trace, a fixed upstream boundary, and fjord outlines on both sides. Since the other three boundaries remain constant, changes in polygon mass serve as a proxy for terminus position change. By comparing these panels, we can visually assess how changes at the terminus influence both discharge and terminus ablation.

For example, we can see a multi-year variability in terminus ablation and discharge for Ullip Sermia (Fig. 7). While the terminus ablation almost matched ice discharge (Fig. 7b), the terminus slightly advanced during 2013–2017 (~1.1 km), indicated by the increase in terminus mass (Fig. 7c).

---

## Author Comment (AC2)

We thank the reviewer for the thoughtful comments and suggestions for the manuscript. Below, we outline our changes for revision based on the comments. The reviewer comments are shown in **blue** and our responses are shown in **black** for clarity.

Summary
This manuscript combines terminus change and ice discharge time series to derive monthly terminus ablation time series for nearly 50 large outlet glaciers distributed across most regions of the Greenland ice sheet. The authors find that most of the sample exhibits coincident seasonal terminus change and ice discharge variability, with a summertime peak in ablation. On seasonal timescales, terminus change contributes to the majority of total ablation, often far exceeding the magnitude of intra-annual variations in flux gate discharge. In light of this, the study concludes that incorporating terminus change is an important component of seasonal and interannual ablation that is excluded from time series using ice discharge time series alone. The manuscript is well written and arranged in a comprehensive and logical structure, with appropriate figures that complement the main results in the text. The methodologies are appropriate for the study and the discussion/conclusions are aligned with the scope of the work presented. This manuscript is therefore nearly suitable for publication in TC in its current form, but there are several aspects of the manuscript that could benefit from additional context and/or clarity, which I detail item by item below:

Main
I think it could be worth including a brief discussion to address types of science questions that can be refined by incorporating a total ablation time series (like the one presented in this study) vs. applications where discharge-only, or similar time series, may be more appropriate. For example, for ensemble mass change studies that often compare Input-Output based methods to altimetry and GRACE, it is useful to derive changed in sea-level contributing mass fluxes. These variations in mass would precede terminus ablation (in conditions where the ablated terminus was floating or near-flotation) because that sea level volume has already been displaced. The manuscript does a good job of describing circumstances (specifically w.r.t fjord conditions and freshening) why total ablation is a refinement over ice flux alone, but does not mention that other mass change related studies may not necessarily benefit from this additional term.

We agree with the reviewer on the relevance of terminus ablation time series and discharge time series for different cases. Based on the suggestion provided, we will add the following paragraph on the discussion on line 299:
*"While the seasonal terminus ablation helps in understanding seasonal processes at the fjord scale, discharge-based time series may be more appropriate for applications focused on large-scale ice sheet mass change and sea-level contribution (Gardner et al. 2013; Jacob et al. 2012). Intercomparisons of GRACE-derived mass loss (Velicogna et al., 2014; Sasgen et al., 2020; Groh et al., 2019) or altimetry-based mass balance (The IMBIE Team) should consider whether discharge or terminus ablation is more directly comparable at regional scales based on the likelihood that glacier termini reach flotation prior to iceberg calving."*

**References:**

Gardner, A. S., Moholdt, G., Cogley, J. G., Wouters, B., Arendt, A. A., Wahr, J., Berthier, E., Hock, R., Pfeffer, W. T., Kaser, G., Ligtenberg, S. R. M., Bolch, T., Sharp, M. J., Hagen, J. O., Van Den Broeke, M. R., & Paul, F. (2013). A reconciled estimate of glacier contributions to sea level rise: 2003 to 2009. Science, 340(6134), 852–857. https://doi.org/10.1126/science.1234532

Groh, A.; Horwath, M.; Horvath, A.; Meister, R.; Sørensen, L.S.; Barletta, V.R.; Forsberg, R.; Wouters, B.; Ditmar, P.; Ran, J.; et al. Evaluating GRACE Mass Change Time Series for the Antarctic and Greenland Ice Sheet—Methods and Results. Geosciences 2019, 9, 415. https://doi.org/10.3390/geosciences9100415

The IMBIE Team. Mass balance of the Greenland Ice Sheet from 1992 to 2018. Nature 579, 233–239 (2020). https://doi.org/10.1038/s41586-019-1855-2

Velicogna, I., Sutterley, T. C., & Van Den Broeke, M. R. (2014). Regional acceleration in ice mass loss from Greenland and Antarctica using GRACE time‑variable gravity data. Geophysical Research Letters, 41(22), 8130–8137. https://doi.org/10.1002/2014gl061052

Sasgen, I., Wouters, B., Gardner, A. S., King, M. D., Tedesco, M., Landerer, F. W., Dahle, C., Save, H., & Fettweis, X. (2020). Return to rapid ice loss in Greenland and record loss in 2019 detected by the GRACE-FO satellites. Communications Earth & Environment, 1(1). https://doi.org/10.1038/s43247-020-0010-1

Jacob, T., Wahr, J., Pfeffer, W. T., & Swenson, S. (2012). Recent contributions of glaciers and ice caps to sea level rise. Nature, 482(7386), 514–518. https://doi.org/10.1038/nature10847

Line 60, On filtering based on BedMachine source: Can the authors provide how many glaciers were excluded due to not meeting the BedMachine source criteria? My understanding was that for the majority of outlets near the margins, mass conservation was a common method for deriving bathymetry estimates (as compared to further inland where kriging is more common). Additionally, how close to the terminus do a direct radar observation hold as applicable to that glacier? For example, do direct observations need to fall within a certain length threshold to be considered robust for the downstream flux ate and terminus thickness calculations?

We selected 58 glaciers across the GrIS based on proximity to radar-based ice thickness/bed elevation estimates. Of these glaciers, 10 did not have sub-annual terminus position time series and were therefore excluded from our seasonal terminus ablation estimates. Most of the glaciers that were excluded due to terminus position availability were above ~80°N (Goliber et al., 2022).
**Reference:**
Goliber, S., Black, T., Catania, G., Lea, J. M., Olsen, H., Cheng, D., Bevan, S., Bjørk, A., Bunce, C., Brough, S., Carr, J. R., Cowton, T., Gardner, A., Fahrner, D., Hill, E., Joughin, I., Korsgaard, N. J., Luckman, A., Moon, T., Murray, T., Sole, A., Wood, M., and Zhang, E.: TermPicks: a century of Greenland glacier terminus data for use in scientific and machine learning applications, The Cryosphere, 16, 3215–3233, https://doi.org/10.5194/tc-16-3215-2022, 2022.

Line 113, Glacier speed-based filtering threshold: How was the 2x averaged speed threshold (used for filtering erroneous terminus advance observations determined? Was this an empirical Threshold?

The choice of the 2x averaged speed threshold was determined empirically as a tradeoff between improving the quality of the time series and minimizing the loss of temporal resolution. We tested different thresholds (0.5x, 1x, 1.5x, 2x, 2.5x, and 3x the averaged speed) across various glaciers to find the optimal balance—eliminating erroneous terminus advance observations while preserving as much of the original dataset as possible. Among these, the 2x averaged speed threshold provided the best results. In the absence of high-spatial-resolution velocity time series, a broader threshold (e.g., 3x maximum flow speed) is recommended (Liu et al. 2021).

**Reference:**
Liu, J., Enderlin, E. M., Marshall, H., & Khalil, A. (2021). Automated detection of marine glacier calving fronts using the 2-D Wavelet Transform Modulus Maxima Segmentation Method. IEEE Transactions on Geoscience and Remote Sensing, 59(11), 9047–9056. https://doi.org/10.1109/tgrs.2021.3053235

Line 150, on unaccounted mass change between terminus and ice flux: I understand that past studies have made similar assumptions given the small overall uncertainty this component would add in to the total ablation. However, for glaciers where persistent retreat occurred throughout the study period, resulting in a terminus much closer to the gate than the beginning of the time period, it could be a useful metric to provide the maximum bias this assumption could possibly impose on the final time series. While the number I likely to be small, providing bounds of uncertainty for at least several glaciers where its impact is likely to be the largest, would help support the decision to exclude mass change over this intermediate region.

Thank you for pointing out that we should minimally provide estimates of mass loss due to surface mass balance from previous studies in our discussion. Based on the suggestion provided, we will add the following paragraph on the discussion on line 291:

….is densely packed within fjords. "*We assume that unaccounted mass change between the terminus and the ice flux gate is minimal due to the close proximity of the flux gates to the termini and the fast flow of these glaciers. The majority of the glaciers in our study exhibit relatively stable inter-annual variability, reducing the likelihood of substantial unaccounted mass loss. Specifically, only 7 glaciers, including the fast-flowing Helheim, Jakobshavn, and Kangerlussuaq glaciers, experienced terminus retreat exceeding 3 km over the 10-year study period (Fig. R1). For these glaciers, ice typically flows at approximately 2 km to 6 km per year (Gardner et al., 2019), facilitating the rapid delivery of ice from the flux gate to the terminus. Even under conditions of elevated melt rates, surface melting over this period would amount to only a few meters of ice thickness. Given that ice thickness at the flux gates generally exceeds several hundred meters, this surface melt contribution represents less than 10% of the total ice thickness, making mass loss due to surface melt between the gate and the terminus a minor component of overall ablation. Kochtitzky et al. (2023) estimate that subaerial melting*

*contributes approximately ~3% to decadal terminus ablation. Similarly, Bollen et al. (2022) report that unaccounted mass change from Greenland's marine-terminating peripheral glaciers amounts to ~0.4 Gt/year relative to a total discharge of ~4 Gt/year. Importantly, these peripheral glaciers flow significantly slower and contribute less discharge than major outlet glaciers. Moreover, estimates for these losses show a decline from ~0.5 Gt/yr when termini were more extended to ~0.3 Gt/yr during the 1999-2018 period. Given these findings, we conclude that the unaccounted mass change between the terminus and the flux gate is within the overall uncertainty range."*

[Figure]

**Fig R1: Terminus position time series for each region, with colors corresponding to the regions in Figure 1.** Each line indicates an individual glacier corresponding to the names in

the lower panel highlighted by the respective region. The numbers in parentheses indicate the number of study glaciers in the region.

**References:**

Bollen, K. E., Enderlin, E. M., and Muhlheim, R.: Dynamic mass loss from Greenland's marine-terminating peripheral glaciers (1985–2018), Journal of Glaciology, pp. 1–11, https://doi.org/10.1017/JOG.2022.52, 2022

Gardner, A. S., Fahnestock, M., and Scambos, T.: MEaSUREs ITS_LIVE Landsat Image-Pair Glacier and Ice Sheet Surface Velocities: Version 1., 2019

Kochtitzky, W., Copland, L., King, M., Hugonnet, R., Jiskoot, H., Morlighem, M., Millan, R., Khan, S. A., & Noël, B. (2023). Closing Greenland's mass balance: frontal ablation of every Greenlandic glacier from 2000 to 2020. Geophysical Research Letters, 50(17). https://doi.org/10.1029/2023gl104095

Line 181, positive mass change from terminus advance: I did not follow the attribution here that negative terminus ablation was due to an underestimation of bias-induced terminus mass loss. Can the authors provide more explanation here? From my understanding, the fact that seasonal signals present in the Fourier analyses necessitate positive terminus change (or "Negative terminus ablation", i.e., advance) in addition to retreat to exhibit seasonal-scale variability. Can the authors clarify whether all instances of terminus advance are considered a result of bias in their analyses, or whether this refers to a specific treatment of terminus change with respect toa. Reference position?

Thank you for calling it to our attention that we need to more clearly explain how to interpret terminus ablation values. As shown in equation 1, change in near-terminus mass change ($\Delta M/\Delta t$) can be positive (when the glacier is advancing), negative (when the glacier is retreating) or zero (no mass loss at the terminus).

$$A_{terminus} = D - (\Delta M/\Delta t),$$

We will add the following section after equation 1 to clarify.

*"The only instance wherein [D – ($\Delta M/\Delta t$)] would be negative is if ($\Delta M/\Delta t$) > 0 (glacier advance) and ($\Delta M/\Delta t$) > D. However, negative terminus ablation doesn't just mean that the terminus is advancing, it means that the terminus is gaining mass at a greater rate than the glacier flow rate. Therefore, we attribute negative terminus ablation largely to interpolation effects."*

Table 2: Consider adding in variance or STD in paratheses beside the mean values for discharge and ablation in each season. This would provide readers with a sense of interannual variability across the regions and how discharge amplitude and seasonality scale with total ablation.

We agree that the inclusion of the standard deviation for each region is helpful for interpretation of the data and we are going to revise the table based on the suggestion as shown below:

**Table 2.** Terminus ablation and discharge (in Gt/yr) values averaged across seasons with standard deviations

| Regions | Winter | | Spring | | Summer | | Fall | |
|---------|-----------|----------|-----------|----------|-----------|----------|-----------|----------|
|         | Discharge | Term Abl | Discharge | Term Abl | Discharge | Term Abl | Discharge | Term Abl |
| NW | 57 (2) | 59 (12) | 58 (2) | 52 (10) | 59 (3) | 85 (28) | 56 (3) | 64 (14) |
| CW | 73 (6) | 72 (9) | 71 (6) | 71 (10) | 74 (6) | 85 (12) | 76 (7) | 70 (10) |
| SW | 8 (0.5) | 8 (0.7) | 8 (0.4) | 9 (1.2) | 8 (0.4) | 9 (0.7) | 7 (0.4) | 7 (0.8) |
| SE | 47 (3) | 42 (15) | 47 (3) | 50 (15) | 47 (3) | 60 (19) | 47 (3) | 41 (19) |
| CE | 41 (2) | 39 (10) | 41 (2) | 30 (13) | 41 (2) | 50 (15) | 40 (2) | 51 (16) |
| NE | 15 (1) | 17 (6) | 15 (1) | 12 (5) | 16 (1) | 24 (16) | 15 (1) | 18 (7) |

Term Abl = Terminus Ablation; Both discharge and terminus ablation are in Gt/yr with values averaged across the entire decade based on the season (rounded to the nearest whole number). Standard deviations are provided in parentheses. Winter = December to February; spring = March to May; summer = June to August; fall = September to November.

---

## Author Response (AR1)

We thank the editor and both reviewers for the thoughtful comments and suggestions for the manuscript. Below, we outline our changes for revision based on the comments. The comments are shown in **blue** and our responses are shown in **black**. Added or modified texts in the manuscript are *italicized* for clarity.

**Review 1: Chad Greene (NASA JPL)**

In this paper, Aman KC and colleagues describe the seasonal advance and retreat of 49 marine terminating glaciers around Greenland. The paper is well written, and the language and figures are polished to perfection. The technical approach employed here is strong and the methods are well described. I know from experience that the task of combining multiple terminus datasets is not an easy one, yet the authors have done an excellent job of making it seem easy, due to the well-reasoned logic they employ and clarity with which they describe it. Overall, the manuscript is of high quality and I believe it will be ready for publication if the results can be framed with a little more context.

The greatest room for improvement lies in the presentation of the current results in relation to previous studies that have conducted similar types of analysis. I would like to see clear and direct statements in the abstract and discussion/conclusion sections that offer insight into how the present results confirm, upends, or reshape our understanding of previous studies. Many previous studies have presented observations of terminus advance and retreat in Greenland, so what makes this one different? How do the current results change the narrative, what's different about this study, what's the big takeaway, and what guidance might the authors give to IMBIE, the IPCC, ice sheet modelers, or whomever might want to build on this work? Do these results support/strengthen our understanding of something that was already known, or introduce something new?

Although the discussion section includes a review of previous studies of physical processes that can cause seasonal advance and retreat, what's missing is a direct comparison of the present results to previous results. Several previous assessments of terminus variability are even cited in this paper, but they are constrained to the Introduction section as a means of providing motivation for the present study. I'd like to see the authors circle back at the end to place the new results in context of previous studies by Black, Joughin, Moon, Kochtitzky, Catania, Fried, Wood, Schild, Cassotto, Howat, Greene, etc. I don't mean to be too prescriptive here, so adjust that list according to taste. Direct quantitative comparisons to all those previous papers may be impossible due to differences in observation periods, but the authors have the intuition and expertise here to guide readers to find key similarities or difference between the present study and previous ones.

Thank you for pointing out that the big take-aways from our analysis are not necessarily clear as is. The main objective of our research is to demonstrate that  dynamic mass loss rates on seasonal scales are strongly influenced by terminus position change even if direct mass loss from terminus retreat is small over decadal time scales. We did not include a discussion of the terminus position change record in the original submission because the focus of the study is on

mass loss whereas the majority of the previous studies of terminus position focused on length or area, not mass, making an intercomparison somewhat difficult. However, based on the reviewer's suggestion, now include an additional figure that highlights seasonal terminus position changes at the study glaciers and the following text in the discussion on line 341:

"…which in turn influence terminus stability. *Different calving styles influence glacier mass balance in distinct ways, with some glaciers experiencing episodic large-scale calving events while others exhibit more gradual, continuous retreat (Cassotto et al., 2015; Catania et al., 2020). While previous analyses of terminus position at seasonal time scales have largely focused on the relationship between terminus change and glacier speed (King et al., 2020; Black and Joughin, 2022, 2023; Moon and Joughin, 2008) , we focus on the direct influence of terminus position change on mass loss. Using terminus position data from previous studies, supplemented with more recent manual delineations created by following the approach recommended by Goliber et al. (2022), we highlight the contribution of seasonal terminus retreat to ice mass loss. For example, in the central west, seasonal terminus position changes ranging from 1500 to 2500 m resulted in an estimated mass loss of approximately 80–100 Gt/year (Fig. 10, S7 and S8)."*

We added the following text in the discussion on line 281:

"… when comparing regional patterns. *In the southwest, terminus ablation tracks the discharge cycle, indicating that smaller serac failure-style calving events remove mass at nearly the same rate as it is delivered to the terminus (Robel, 2017; Fahrner et al., 2021). For the central west, previous studies have shown that both seasonality in ice mélange extent and submarine melting can influence seasonal terminus position change (Moon and Joughin, 2008; Mortensen et al., 2020; Howat et al., 2010). However, their relative importance varies with glacier geometry, driving widespread terminus retreat (Greene et al., 2024; Fried et al., 2018) and enhanced terminus ablation throughout the first half of the melt season that peaks in July (Rignot et al., 2016; Carroll et al., 2015, 2016). As with the central west sector, seasonal retreat of glaciers in the northwest has been in part attributed to the break-up of sea ice and ice mélange (Carr et al., 2013) and our terminus ablation time series indicate that the near-coincident retreat of glaciers after an early-summer maximum extent causes a large seasonal spike in terminus ablation (Greene et al., 2024) (Fig. 10)."*

*We added the following text in the discussion on line 307:*

"….decadal terminus ablation (Kochtitzky et al., 2023). *When averaged over the 2013 -2023 study period, our terminus ablation estimates are in good agreement with the decadal-scale estimates of (Kochtitzky et al., 2023)(Fig. S9). Differences between these terminus ablation estimates are likely due to variations in the temporal resolution of the terminus position and discharge time series (e.g., monthly vs. decadal), the source of discharge data data (Mankoff et al., 2021; Kochtitzky and Copland, 2022), and the treatment of ice thickness (single value vs. dynamic) (Fahrner et al., 2025).* These differences exceed the adjustments made for surface mass balance. Further support for the relatively…"

We also included the following plots for each region in the supplement that show the seasonal retreat as a direct contributor to mass loss.

[Figure]

Figure S7. Seasonality of terminus position time series for east Greenland, with colors corresponding to the regions in Fig. 1. Each line indicates an individual glacier, corresponding to the names in the side panel — which groups glaciers by region. The numbers in parentheses indicate the number of study glaciers in the region.

[Figure]

Figure S8. Seasonality of terminus position time series for west Greenland, with colors corresponding to the regions in Fig. 1. Each line indicates an individual glacier, corresponding to the names in the side panel — which groups glaciers by region. The numbers in parentheses indicate the number of study glaciers in the region.

[Figure]

Figure S10. terminus position time series for east Greenland, with colors corresponding to the regions in Fig. 1. Each line indicates an individual glacier, corresponding to the names in the side panel — which groups glaciers by region. The numbers in parentheses indicate the number of study glaciers in the region.

[Figure]

Figure S11. terminus position time series for west Greenland, with colors corresponding to the regions in Fig. 1. Each line indicates an individual glacier, corresponding to the names in the side panel — which groups glaciers by region. The numbers in parentheses indicate the number of study glaciers in the region.

Note: The data and code availability statements in this paper point to links that do not currently exist. Accordingly, this review does not apply to the data or code, and I'm taking it on blind faith that the data is of adequate quality and will be made available as promised.

The correct links have been updated.

Minor comments:
Line 11 Regarding this sentence in the abstract:
*"for the northwest and central west sectors, where the fraction of outlet glaciers included in our estimates is greatest, the average difference between the annual maximum and minimum in terminus ablation are ~51Gt/yr and ~25Gt/yr, respectively, compared to only ~5Gt/yr for discharge."*

That's really hard to parse, especially for someone who has not yet read the paper. I recommend rewording to reduce the number of comma-bound clauses, because they tend to break the flow and force the reader to mentally keep track of too many little concepts and their relationships to each other.

The sentence also requires the reader to have a prior understanding of what is meant by the somewhat ambiguous phrase "annual maximum and minimum terminus ablation", which I can

try to guess the meaning of, but not confidently enough to understand the significance of its quantified rates of 51 Gt/yr vs 25 Gt/yr.

Thank you for the helpful feedback. We have modified the abstract for clarity:
*"However, seasonal variations in terminus ablation are much larger than those in discharge and are the primary driver of seasonality in terminus ablation. For the northwest and central west sectors, where the highest fraction of outlet glaciers is included in our terminus ablation dataset, terminus ablation varies by ~51 Gt/yr and ~25 Gt/yr, respectively, over each year. In contrast, the corresponding variation in discharge is only ~5 Gt/yr."*

Line 115 and elsewhere I think AERODEM should be written in all caps?
https://www.nodc.noaa.gov/archive/arc0088/0145405/1.1/data/0-data/G150AERODEM/

Thank you for the suggestion. We have changed "AeroDEM" to "AERODEM".

Figure 4 is very nice. I appreciate how the simplicity of the diagram focuses attention on the issues that can cause anomalous terminus position picks, so the figure will be useful to anyone who has not encountered these issues firsthand.

We are glad that you found the simplicity of Figure 4 effective in highlighting the key issues.

**Line 157-168** The equations in Sec 2.2.1 are relatively innocuous, but I think they're relatively standard mathematical formulations, right? If there's something nonstandard going on here, be sure to mention it explicitly, otherwise I think it would be sufficient to say you plot the power spectral density of the terminus ablation time series and remove the equations. Rather than reading equations, in this section I would like to see PSD plot(s) that illustrate seasonal vs erratic glaciers because a major conclusion of the paper depends entirely on how the PSD is interpreted. Showing those PSD plots will help readers gain an intuition for how the results are obtained and how sensitive the overall findings might be to subjective differences in interpretation.

We agree that the equations presented are standard mathematical formulations for calculating the power spectral density of the terminus ablation time series and we will remove them. Regarding your suggestion to include PSD plots, we initially explored this approach. However, the periodogram values exhibit considerable variability across different glaciers, making it challenging to present a clear visual distinction between seasonal and erratic behaviors through plots alone (shown below).

[Figure]

This variability can obscure meaningful patterns and increase the likelihood of subjective interpretation. Given these challenges, we chose to present the classification in tabular form, as it more concisely and objectively conveys the primary results without relying on interpretative nuances from the PSD plots.

We included the following table in the supplement after formatting it to meet the journal's requirements:

**Table 2.** Glacier properties including frequency, location, and seasonality classification

| Glacier | Region | Frequency | Latitude | Longitude | Seasonality | Long-term Interannual |
|---|---|---|---|---|---|---|
| Sermeq_Avannerleq | CW | 12.10 | 70.0853 | -50.2468 | C | N |
| Innaqsisoqssup_Oqquani_Sermeq | NW | 4.52 | 76.3833 | -62.7665 | E | E |
| Sermeq_Avannarleq | NW | 12.10 | 73.9410 | -55.7679 | C | N |
| Helheim_Gletsjer | SE | 12.10 | 66.3735 | -38.3067 | C | N |
| Tuttulipaluuup_Sermia | NW | 12.10 | 77.7000 | -66.2330 | C | N |
| Kangerdluusuaq_Sermia | CW | 12.10 | 71.4588 | -51.3101 | C | N |
| Kangerdluusuaq_Gletsjer | CE | 12.20 | 68.6746 | -33.0760 | C | N |
| Qaqujaarsuup_Sermia | NW | 5.76 | 77.5167 | -65.6664 | C | N |
| Apuseerajik | SE | 12.10 | 66.3820 | -37.5439 | C | N |
| Nordenskiold_Gletsjer | NW | 60.50 | 75.8248 | -59.0399 | C | Y |
| Zachariae_Isstrom | NE | 11.00 | 78.9333 | -21.0000 | C | N |
| Sermeq_Kujalleq_N1 | CW | 12.20 | 69.9960 | -50.1596 | C | N |
| Sermeq_Silarleq | CW | 12.20 | 70.8268 | -50.7627 | C | N |
| Sermersuaq | NW | 3.27 | 79.4462 | -63.3957 | E | N |
| Tuttuliksasap_Sermia | NW | 12.10 | 74.9618 | -57.0355 | C | N |
| Nansen_Gletsjer | NW | 6.05 | 75.7759 | -58.8283 | E | Y |
| Sermeq_Kujalleq_N3 | NW | 12.10 | 73.8317 | -55.5825 | C | N |
| Sermilik | CW | 12.10 | 70.6333 | -50.6167 | C | N |
| Ikissuup_Sermersua | NW | 12.78 | 74.2307 | -55.8275 | C | N |
| Christian_IV_Gletsjer | CE | 12.20 | 68.7000 | -30.6167 | C | N |
| Qeqqertasuusaq_Sermia | NW | 12.20 | 77.6564 | -65.9679 | C | N |
| Sermeq_Avannarleq_N2 | CW | 4.46 | 70.5454 | -50.4896 | E | N |
| Kangilernata_Sermia | CW | 12.10 | 69.9000 | -50.3420 | C | N |
| Qeqqertasuup_Sermia | NW | 6.05 | 73.5932 | -55.5306 | C | N |
| Rimfaxe | SE | 12.20 | 63.3050 | -42.3669 | C | N |
| Kangerdluarsup_Sermia | NW | 12.60 | 77.6934 | -68.5829 | C | N |
| Narsap_Sermia | SW | 12.10 | 64.6672 | -49.8576 | C | N |
| Perlerfiup_Sermia | CW | 12.89 | 70.9090 | -50.9227 | C | N |
| Bernstorff | SE | 12.10 | 63.8846 | -41.7654 | C | N |
| Saqqarliup_Sermia | CW | 11.00 | 68.8667 | -50.2833 | C | N |
| Akullersuup_Sermia | SW | 12.20 | 64.3833 | -49.4779 | C | N |
| Apusiigajik | SE | 12.20 | 63.2926 | -41.9168 | C | N |
| Qeqertat_Timaanni_Sermeq | NW | 8.64 | 76.3000 | -61.7666 | C | N |
| Yngvar_Nielsen_Gletsjer | NW | 11.00 | 76.3334 | -64.0831 | E | N |
| Ullip_Sermia | NW | 12.20 | 76.5797 | -67.6260 | C | N |
| Sermeq_Kujalleq | CW | 5.81 | 69.1833 | -49.8000 | E | Y |
| Naajarsuit_Sermiat | NW | 12.10 | 73.2500 | -55.0833 | C | N |
| Heimdal | SE | 12.20 | 62.9052 | -42.6851 | C | N |
| Graulv | SE | 12.10 | 64.3500 | -41.5667 | C | N |
| Frederiksborg_Gletsjer | CE | 11.60 | 68.4557 | -31.6620 | C | N |
| Illullip_Sermia | NW | 12.10 | 74.4167 | -55.9666 | C | N |
| Rink_Gletsier_N | NW | 5.76 | 76.2167 | -60.9999 | C | N |
| Issuusarsuit_Sermiat | NW | 12.20 | 76.0667 | -60.6333 | C | Y |
| Daugaard_Jensen_Gletsjer | CE | 11.50 | 71.7500 | -29.0000 | C | N |
| Sermeq_Kujalleq_N2 | CW | 2.52 | 70.4054 | -50.5364 | E | N |
| Dietrichson_Gletsjer | NW | 12.30 | 75.4582 | -58.0637 | C | N |
| Kangiata_Nunaata_Sermia | SW | 4.01 | 64.2966 | -49.6102 | E | N |
| Sverdrup_Gletsjer | NW | 11.60 | 75.6195 | -57.9704 | C | N |
| Eqip_Sermia | CW | 12.20 | 69.8080 | -50.1851 | C | N |

C = Consistent seasonal variability; E = Erratic; N = No long-term variability; Y = Present long-term variability. Frequency is given in cycles per year, rounded to two decimal places.

**Figures 5-7** I think I'm missing something here, because panel c is presented at monthly resolution, whereas the observations in panel b are presented in irregular intervals. I would

expect the overall Mass time series to be the integral of the Ice Flux time series, but that's not what these panels look like. Please explain how they're related to each other.

In Figures 5-7, the relationship between panels b and c lies in how terminus mass change and ice flux are derived and processed.
Panel c presents ice flux components in a standardized monthly time series. Because these mass fluxes are from different datasets with varying temporal resolutions, they are either linearly interpolated or weighted to ensure comparability at a monthly scale.

Panel b represents near-terminus mass changes, which are based on observed terminus position data and are often irregular in time. This panel highlights the uncertainty in total mass estimates, which are primarily from terminus position records. The terminus polygons used for mass estimation are constructed using each available terminus trace, a fixed upstream boundary, and fjord outlines on both sides. Since the other three boundaries remain constant, changes in polygon mass serve as a proxy for terminus position change. By comparing these panels, we can visually assess how changes at the terminus influence both discharge and terminus ablation.

For example, we can see a multi-year variability in terminus ablation and discharge for Ullip Sermia (Fig. 7). While the terminus ablation almost matched ice discharge (Fig. 7b), the terminus slightly advanced during 2013–2017 (~1.1 km), indicated by the increase in terminus mass (Fig. 7c).

**Review 2:**

Summary
This manuscript combines terminus change and ice discharge time series to derive monthly terminus ablation time series for nearly 50 large outlet glaciers distributed across most regions of the Greenland ice sheet. The authors find that most of the sample exhibits coincident seasonal terminus change and ice discharge variability, with a summertime peak in ablation. On seasonal timescales, terminus change contributes to the majority of total ablation, often far exceeding the magnitude of intra-annual variations in flux gate discharge. In light of this, the study concludes that incorporating terminus change is an important component of seasonal and interannual ablation that is excluded from time series using ice discharge time series alone. The manuscript is well written and arranged in a comprehensive and logical structure, with appropriate figures that complement the main results in the text. The methodologies are appropriate for the study and the discussion/conclusions are aligned with the scope of the work presented. This manuscript is therefore nearly suitable for publication in TC in its current form, but there are several aspects of the manuscript that could benefit from additional context and/or clarity, which I detail item by item below:
Main
I think it could be worth including a brief discussion to address types of science questions that can be refined by incorporating a total ablation time series (like the one presented in this study) vs. applications where discharge-only, or similar time series, may be more appropriate. For

example, for ensemble mass change studies that often compare Input-Output based methods to altimetry and GRACE, it is useful to derive changed in sea-level contributing mass fluxes. These variations in mass would precede terminus ablation (in conditions where the ablated terminus was floating or near-flotation) because that sea level volume has already been displaced. The manuscript does a good job of describing circumstances (specifically w.r.t fjord conditions and freshening) why total ablation is a refinement over ice flux alone, but does not mention that other mass change related studies may not necessarily benefit from this additional term.

We agree with the reviewer that terminus ablation may not always be the most appropriate dataset to use for different types of data intercomparisons. Based on the suggestion provided, we added the following paragraph in the discussion on line 317:

*Biases in bed elevation near glacier termini that are in excess of the quoted uncertainties in BedMachine may influence our terminus ablation estimates but cannot be constrained with observations. Biases in bed elevation are particularly important for termini near flotation because they could cause floating ice to be falsely mapped as grounded, resulting in ice thickness over-estimation. The identification of floating ice is also dependent on the accuracy of density estimates. Here we estimate terminus mass using a slightly smaller density ($917 \text{ kg m}^{-3}$) by Mankoff et al. (2021) than used to calculate discharge ($917 \text{ kg m}^{-3}$) in order to account for the influence of crevassing on near-terminus ice density. Although the differences in density between datasets (<2%) should have a relatively small influence on terminus ablation estimates, they can influence the identification of floating ice. The amplitude of seasonal oscillations in terminus ablation rate will be exaggerated where floating tongues seasonally form and disintegrate, but most glaciers in Greenland no longer maintain perennial ice tongues (Enderlin and Howat, 2013; Catania et al., 2020) and we expect seasonal floating tongues to be fairly short. Given uncertainties in the extent of floating tongues, intercomparisons among GRACE-derived mass loss (e.g., Velicogna et al. (2020); Sasgen et al. (2020)), altimetry (Shepherd et al., 2019; Felikson et al., 2017), and input-output methods (Mouginot et al., 2019; Colgan et al., 2019) should continue to use discharge, which reflects the mass flux across the gate. For grounded termini, the impact on sea-level is governed by the fraction of ice above flotation, since ice already displacing water does not contribute to sea-level rise upon calving. In such contexts, incorporating total ablation may overestimate the sea-level relevant mass loss unless the floating fraction is taken into account. For analysis of seasonal processes at the fjord scale, however, the terminus ablation time series are more appropriate even when considering their uncertainties."*

Line 60, On filtering based on BedMachine source: Can the authors provide how many glaciers were excluded due to not meeting the BedMachine source criteria? My understanding was that for the majority of outlets near the margins, mass conservation was a common method for deriving bathymetry estimates (as compared to further inland where kriging is more common). Additionally, how close to the terminus do a direct radar observation hold as applicable to that glacier? For example, do direct observations need to fall within a certain length threshold to be considered robust for the downstream flux ate and terminus thickness calculations?

We selected 58 glaciers across the GrIS based on proximity to radar-based ice thickness/bed elevation estimates. Of these glaciers, 10 did not have sub-annual terminus position time series and were therefore excluded from our seasonal terminus ablation estimates. Most of the glaciers that were excluded due to terminus position availability were above ~80°N (Goliber et al., 2022).

Line 113, Glacier speed-based filtering threshold: How was the 2x averaged speed threshold (used for filtering erroneous terminus advance observations determined? Was this an empirical Threshold?

The choice of the 2x averaged speed threshold was determined empirically as a tradeoff between improving the quality of the time series and minimizing the loss of temporal resolution. We tested different thresholds (0.5x, 1x, 1.5x, 2x, 2.5x, and 3x the averaged speed) across various glaciers to find the optimal balance — eliminating erroneous terminus advance observations while preserving as much of the original dataset as possible. Among these, the 2x averaged speed threshold provided the best results. In the absence of high-spatial-resolution velocity time series, a broader threshold (e.g., 3x maximum flow speed) is recommended (Liu et al. 2021).

Line 150, on unaccounted mass change between terminus and ice flux: I understand that past studies have made similar assumptions given the small overall uncertainty this component would add into the total ablation. However, for glaciers where persistent retreat occurred throughout the study period, resulting in a terminus much closer to the gate than the beginning of the time period, it could be a useful metric to provide the maximum bias this assumption could possibly impose on the final time series. While the number is likely to be small, providing bounds of uncertainty for at least several glaciers where its impact is likely to be the largest, would help support the decision to exclude mass change over this intermediate region.

We agree that large decadal-scale retreat could influence the magnitude of surface mass loss between the discharge gate and the terminus and we looked into total retreat over our observation period as a result. We have added some statistics on total retreat to support our assumption that surface mass loss between the gate and the terminus is relatively small over the full study period and provided estimates of surface mass loss between discharge gates and termini from other studies to support our assumption in the following paragraph on the discussion (starting on line 296):

"negative terminus ablation estimates occur (Fig. S6). *Our assumption that mass change between the terminus and the flux gate due to surface accumulation and ablation is minimal may slightly bias terminus ablation estimates, but it is unlikely to cause negative terminus ablation rates because the close proximity of the flux gate and terminus and fast ice flow minimizes the travel time to the terminus. Since terminus position varies on seasonal to decadal time scales, surface mass balance changes between the flux gate and terminus could potentially bias terminus ablation estimates. However, seasonal terminus retreat is relatively*

*small (~1 km) and only 7 glaciers retreated >3 km from 2013-2023 (Fig. S10 and S11). For the glaciers with the largest magnitude of retreat, surface speeds are ~2-6 km per year (Gardner et al., 2019) such that the maximum surface mass balance correction between the flux gate and the most extended terminus position would only be on the order of a few meters of ice thickness. Given that ice thickness at the flux gates generally exceeds several hundred meters, the surface mass balance adjustment would be <<10 % of the total ice thickness throughout the entire study period. An independent estimate of surface mass balance between flux gates and termini for 213 of Greenland's outlet glaciers suggests it contributes only ~3% to decadal terminus ablation (Kochtitzky et al., 2023). When averaged over the 2013 -2023 study period, our terminus ablation estimates are in good agreement with the decadal-scale estimates of (Kochtitzky et al., 2023)(Fig. S9). Differences between these terminus ablation estimates are likely due to variations in the temporal resolution of the terminus position and discharge time series (e.g., monthly vs. decadal), the source of discharge data data (Mankoff et al., 2021; Kochtitzky and Copland, 2022), and the treatment of ice thickness (single value vs. dynamic) (Fahrner et al., 2025). These differences exceed the adjustments made for surface mass balance. Further support for the relatively small mass change caused of surface accumulation and ablation between the flux gates and termini comes from Greenland's peripheral glaciers, which are generally slower-flowing than outlet glaciers; for peripheral glacier, surface mass balance adjustments decreased by 0.2 Gt/yr from 1999-2018 due to terminus retreat and were only ~10% of discharge on average (Bollen et al., 2022). Therefore, we conclude that surface mass change between the terminus and the flux gate is within the overall uncertainty range."*

Line 181, positive mass change from terminus advance: I did not follow the attribution here that negative terminus ablation was due to an underestimation of bias-induced terminus mass loss. Can the authors provide more explanation here? From my understanding, the fact that seasonal signals present in the Fourier analyses necessitate positive terminus change (or "Negative terminus ablation", i.e., advance) in addition to retreat to exhibit seasonal-scale variability. Can the authors clarify whether all instances of terminus advance are considered a result of bias in their analyses, or whether this refers to a specific treatment of terminus change with respect toa. Reference position?

Thank you for calling it to our attention that we need to more clearly explain how to interpret terminus ablation values. As shown in Equation 1, change in near-terminus mass change ( $\Delta M/\Delta t$) can be positive (when the glacier is advancing), negative (when the glacier is retreating) or zero (no mass loss at the terminus).

$$A_{terminus} = D - (\Delta M/\Delta t),$$

We will add the following section after Equation 1 to clarify.

*"The only instance wherein [D – ($\Delta M/\Delta t$)] would be negative is if ($\Delta M/\Delta t$) > 0 (glacier advance) and ($\Delta M/\Delta t$) > D, suggesting that the terminus is gaining mass at a greater rate than is provided by glacier flow. Since ice is not accreting along the terminus face, we attribute negative terminus ablation largely to interpolation effects."*

Table 2: Consider adding in variance or STD in paratheses beside the mean values for discharge and ablation in each season. This would provide readers with a sense of interannual variability across the regions and how discharge amplitude and seasonality scale with total ablation.

We agree that the inclusion of the standard deviation for each region is helpful for interpretation of the data and we revised the table based on the suggestion, as shown below:

**Table 2.** Terminus ablation and discharge (in Gt/yr) values averaged across seasons with standard deviations

| Regions | Winter | | Spring | | Summer | | Fall | |
|---|---|---|---|---|---|---|---|---|
| | Discharge | Term Abl | Discharge | Term Abl | Discharge | Term Abl | Discharge | Term Abl |
| NW | 57 (2) | 59 (12) | 58 (2) | 52 (10) | 59 (3) | 85 (28) | 56 (3) | 64 (14) |
| CW | 73 (6) | 72 (9) | 71 (6) | 71 (10) | 74 (6) | 85 (12) | 76 (7) | 70 (10) |
| SW | 8 (0.5) | 8 (0.7) | 8 (0.4) | 9 (1.2) | 8 (0.4) | 9 (0.7) | 7 (0.4) | 7 (0.8) |
| SE | 47 (3) | 42 (15) | 47 (3) | 50 (15) | 47 (3) | 60 (19) | 47 (3) | 41 (19) |
| CE | 41 (2) | 39 (10) | 41 (2) | 30 (13) | 41 (2) | 50 (15) | 40 (2) | 51 (16) |
| NE | 15 (1) | 17 (6) | 15 (1) | 12 (5) | 16 (1) | 24 (16) | 15 (1) | 18 (7) |

Term Abl = Terminus Ablation; Both discharge and terminus ablation are in Gt/yr with values averaged across the entire decade based on the season (rounded to the nearest whole number). Standard deviations are provided in parentheses. Winter = December to February; spring = March to May; summer = June to August; fall = September to November.